

# Return period of high-dimensional compound events. Part I: Conceptual framework

Manuel Del Jesus[1], Diego Urrea Méndez[1], and Dina Vanessa Gomez Rave[1]

[1]IHCantabria - Instituto de Hidráulica Ambiental de la Universidad de Cantabria, Santander, Spain.

**Correspondence:** Manuel Del Jesus (manuel.deljesus@unican.es)

**Abstract.** Natural hazards like floods and droughts result from the complex interplay of multiple physical processes across various spatial and temporal scales. Traditional univariate analyses fall short in capturing the devastating impacts of compound events—where multiple drivers interact, often leading to more severe outcomes. This study expands on existing methodologies to quantify compound events by introducing a robust framework that integrates hydrological, statistical, and machine learning

techniques. We propose a novel approach for defining the critical layer associated with multivariate return periods in higher-dimensional spaces, addressing the challenges in modeling interactions beyond two or three variables. This research not only enhances the understanding of compound events but also provides practical tools for their analysis, offering significant implications for climate risk assessment and environmental management. A forthcoming second paper will demonstrate the practical application of this methodology, focusing on calculating the multivariate return period in five dimensions for rainfall

dependencies across different locations.

## 1 Introduction

Natural hazards such as floods and droughts arise from the interaction of multiple physical processes at different spatial and temporal scales (Blöschl et al., 2020; Kundzewicz et al., 2014; Van Loon et al., 2016). Flooding, for instance, may be induced by the interaction between precipitation and soil moisture content, by the combined contribution from rainfall at different

locations, or even by the combination of intense rainfall events over several days. Therefore, the analysis of the magnitude of extreme impacts should account for the co-occurrence of different processes (or the same process at different locations in space and/or moments in time). This combined characterization is essential to derive accurate representations of the extremes, which becomes specially important when dealing with hydrologic and hydraulic design values. Traditionally, practitioners have focused on the use of univariate analysis, which considers only individual events or factors (Wilks, 2006). However, numerous

studies have shown that the interaction of multiple drivers or events, especially across various locations, can lead to far more devastating impacts than when these factors are considered in isolation (Zscheischler et al., 2018, 2020).

Understanding the complex interactions among drivers and physical processes is fundamental for evaluating and managing resulting risks. Such combined events, involving the simultaneous occurrence and interaction of different processes, are commonly referred to as *Compound Events*. In its latest report (IPCC, 2023), the Intergovernmental Panel on Climate Change

(IPCC) defines compound events (so-called 'compound extremes' and 'compound extreme events') as the combination of





multiple drivers and/or hazards that contribute to social and/or environmental risk. This definition, introduced in the Special Report on Climate Extremes (IPCC et al., 2012), highlights the importance for researchers and society to quantify and understand the spatial and/or temporal dependence among drivers. It is likely that global warming will increase the probability of compound events in many regions, representing an increase in the frequency of droughts, floods, and wildfires (IPCC, 2023).

The scientific community has made significant efforts to understand and provide methodologies for the study of compound events. Initiatives such as the European project DAMOCLES have been developed to better understand, describe, and project compound events. In the literature, different studies can be found, such as that of Le et al. (2018), who proposed the use of influence diagrams for defining, mapping, analyzing, modeling, and communicating the risk of compound events. Additionally, Bensi et al. (2022) present a critical review of current practices, approaches, and example use case studies for flood risk

assessment. Zscheischler et al. (2020) propose a framework for the analysis, classification, and modeling of compound events and Zscheischler et al. (2022) present case studies and include a literature review of various articles that contain the latest advances for the study of compound events.

    Although the proposed methodological frameworks for analyzing compound events are relatively new, many of their components are based on statistical and mathematical methods developed over decades. For example, Zscheischler et al. (2020) proposes

three steps for analyzing compound events. The first, *Diagnosis of compound-event drivers*, describes how defining the drivers and selecting the appropriate temporal and spatial scales is key to identifying the drivers of hazards. The second step, *Quantifying compound effects*, highlights the importance of identifying the strength of the relationships between different causal components. This step, for example, relies on the application of classical statistical methods that allow quantifying and modeling the relationships between variables (correlation coefficients, copulas, Poisson processes, etc.). The third step,

*Mapping drivers on impacts*, focuses on defining how the combination of different events can cause a specific impact. This is done using *AND* scenarios, where all events must occur together, and *OR* scenarios, where only one of the events needs to occur. The frequency of these event combinations, can be estimated by calculating the multivariate return period.

    The correlation coefficient, copulas, and multivariate return period are fundamental elements for analyzing compound events. These concepts have been extensively studied and applied in various fields, including hydrology (Salvadori, 2004; Grimaldi

and Serinaldi, 2006; Volpi and Fiori, 2012; Brunner et al., 2016; Tosunoglu et al., 2020). Copulas play an essential role in analyzing phenomena such as droughts, storms, and floods because they capture the dependence structure between variables and can model their joint behavior. Their use in hydrology has focused on applications where only two, or at most three processes were involved. Most applications focus on synthetic event generation and modeling for determining Joint Return Periods (JRPs). Recently, the use of vine copulas has increased for exploring multidimensional spaces (Gräler, 2014; Nguyen-

Huy et al., 2017; Zhang et al., 2022), demonstrating great potential in hydrological applications. Vine copulas are a flexible class of dependence models consisting of bivariate building blocks. Joe (1996) proposed the construction of the first pair of copulas of a multivariate copula using distribution functions, while Bedford and Cooke (2001) and Bedford and Cooke (2002) developed independent approaches expressed in terms of densities. For more theoretical details, please refer to (Sklar, 1959; Nelsen, 2006).





In the multivariate case, the concept of return period can be redefined using copulas, since they can represent the multivariate cumulative distribution function. The development of this concept has been notable in recent decades (Favre et al., 2004; Salvadori, 2004; De Michele et al., 2007; Salvadori and De Michele, 2010; Salvadori et al., 2011; Gräler, 2014; Gräler et al., 2016). Numerous advances have been made in multivariate frequency analysis in hydrology; however, much of the literature has focused on bivariate cases (Zscheischler et al., 2020) and, in a few cases, in trivariate spaces (Grimaldi and Serinaldi,

2006; Pinya et al., 2009; Muthuvel and Mahesha, 2021). Although part of the literature mentions that copula theory and multivariate return periods for the bi- and trivariate cases (2 and 3 dimensions, respectively) are applicable in the n-variate case (n dimensions) (Brunner et al., 2016), the mathematical formulation becomes complicated as the number of variables increases (Grimaldi and Serinaldi, 2006). Choosing the appropriate temporal and spatial scales for an event of interest is a challenge and becomes even more difficult in higher dimensions (Zscheischler et al., 2020). Obtaining a joint cumulative distribution

function from a vine copula is a challenge from a theoretical standpoint since there is no general exact solution that is also computationally efficient (Vernieuwe et al., 2015). Due to the described challenges, there is a clear lack of studies addressing the multivariate return period of compound extreme events in dimensional spaces exceeding three dimensions.

   In this context, this article has two main objectives: (I) To complement the existing methodological framework associated with compound events and JRPs, providing practical guidelines and detailed steps specifically addressing events involving two

or more drivers. (II) To propose a methodological approach that coherently integrates hydrological, statistical, mathematical, and machine learning concepts to address the challenges associated with JRP analysis considering more than two drivers. Specifically, this latter objective will focus on a methodological framework for defining the critical layer or hypersurface associated with the multivariate return period in n dimensions, also considering a computationally efficient method. A second part of this article will apply the proposed methodology to a real case study, focusing on the spatial dependence of rainfall

measured at different locations.

## 2   Evaluation of copula families for modeling dependencies in compound events

In the literature, various families of copula functions can be found. Each of them is better suited to capture different types of dependencies among variables, offering dependence structures, symmetries, and tail behaviors that, collectively, provide great flexibility for modeling compound events. The selection of the copula family and type for modeling tail dependence is crucial

when dealing with compound events (Zscheischler et al., 2020). The copula families typically used in climate and hydrological studies are Meta-Elliptical, Archimedean, and Extreme-Value (Chen and Guo, 2019). In this section, we provide a description of the applicability of each of these copula families. We also discuss which are useful for modeling compound events that show dependencies in two or more dimensions, while also considering tail dependencies.

   Meta-Elliptical copulas, such as Gaussian and t-Student, are commonly applied to model dependencies in dimensions

greater than two (Genest et al., 2007; Hao et al., 2017); however, Gaussian copulas are less suitable for modeling compound events with tail dependencies due to their limited ability to capture such relationships (Jaser and Min, 2021). Archimedean copulas are notable for their flexibility in capturing both lower and upper tail dependencies (Nelsen, 2006), making them





particularly useful for compound events like droughts or floods (Genest and Favre, 2007). Extreme-Value copulas are ideal for modeling dependence among compound events with tail dependencies and are a convenient choice when analyzing generally

positive dependence structures (Gudendorf and Segers, 2010). On the other hand, Vine copulas not only offer great flexibility by allowing the modeling of complex dependencies in high-dimensional spaces, but they are also capable of capturing tail dependencies in these contexts (Czado, 2019). Each of these families has its specific application in the modeling of compound events, underscoring the importance of selecting the appropriate copula based on the type of dependency and the tail behavior of the involved marginal distributions.

In addition to these families, other theoretical copulas may also be useful in multivariate analysis, depending on the specific characteristics of the variables under study. For a more detailed description and a deeper understanding of the mathematical framework and corresponding equations for all the copula families mentioned, it is recommended to consult the supplementary information. In that section, the discussion on the applicability and properties of each copula is considerably expanded, providing practical examples and an exhaustive analysis that complements the information presented here.

## 3    Description of the methodological framework

To achieve the proposed objectives, we introduce a six-step methodological framework. The first step involves defining the drivers, selecting appropriate temporal and spatial scales, and identifying the drivers of hazards, following the methodological framework developed by (Zscheischler et al., 2020). The second step, termed Informative Exploratory Data Analysis, focuses on data preprocessing and conducting a comprehensive dependency analysis. The third step involves testing different structures and models, specifically tree structures and copulas, to identify those with the best performance. This is accomplished through the application of statistical methods, including tail dependence tests and goodness-of-fit tests. In the fourth step, hazard scenarios are defined, described in the literature as return periods in a multidimensional context, with the aim of providing a general framework for applying this concept to multidimensional spaces. In the fifth step, we present the methodological approach for defining the critical layer for 2, 3, and $n$ dimensions, addressing the challenges that arise with increasing dimensionality. The critical layer defines all events corresponding to the selected JRP. Finally, in the sixth step, we propose a methodology to derive design events based on the conceptual framework outlined in step 5.

Steps 1 to 4 provide a description of the current state of the art, offering practical guidelines commonly applied when dealing with compound events involving two or more drivers. Steps 5 and 6 introduce a new methodological approach that integrates hydrological, statistical, mathematical, and machine learning concepts to address the challenges associated with JRPs analysis in a multidimensional or multispacial context. The complete methodological framework is presented in Fig. 1.

### 3.1    Diagnostics

In the analysis of compound events, it is crucial to understand the underlying phenomena and identify the drivers associated with the hazards that could generate an impact. This first step, which corresponds to the diagnostic stage described in Fig. 1, involves acquiring a deep understanding of the different hazards that could trigger a compound event and determining which




**Figure 1.** Proposed methodological framework. The figure outlines a six-step process for analyzing compound events. Step 1 involves the Diagnosis of the event, including typology, drivers, and scale definition. Step 2 focuses on the Quantification of Compound Events through exploratory dependency analysis and data pre-treatment. Step 3 addresses Dependence Structures, followed by Step 4, which defines Hazard Scenarios. Step 5 introduces the Definition of Multi-Dimensional Critical Layer, and Step 6 presents the approach for identifying Multivariate Design Events with the highest probability density, providing a set of events associated with the same return period.





factors or variables influence them. During the diagnostic stage, drivers must be selected by identifying their typology and defining the spatial and temporal scale of the study.

According to Zscheischler et al. (2020), compound weather events can be typologically classified into four categories: preconditioned, where a pre-existing climate-driven condition can cause an impact; multivariate, where the combination of multiple drivers leads to an impact; temporally compounding, where the succession of hazards leads to an impact; and spatially

compounding, where hazards in multiple connected locations cause an aggregated impact. Given the inherent complexity of compound events, it is important to recognize that in certain cases, they may fit into more than one category, implying that the boundaries within the typology must be flexible. This flexibility is necessary because compound events can exhibit unique characteristics and combinations that challenge a rigid classification.

By obtaining an initial understanding of the underlying phenomena and their typology, a solid starting point is established for

analyzing and evaluating compound events. This allows for a better understanding of the causal mechanisms and interactions between the various elements of the system. For more information, refer to Zscheischler et al. (2020).

## 3.2  Quantification of compound events

In this chapter, we delve into the critical process of quantifying compound events, a multifaceted approach essential for understanding the complex interdependencies between variables in environmental and climate studies. As outlined in Figure 2,

the chapter is structured around three primary components: 2A. Exploratory Dependency Analysis, 2B. Pre-Treatment of Data, and 2C. Marginal Models. These steps are integral to ensuring that the data is not only accurately analyzed but also adequately prepared for further modeling. It is important to note that this figure also provides a more detailed explanation of Step 2 from Fig. 1, offering a deeper insight into the processes involved.

### 3.2.1  Exploratory dependency analysis

**Measuring Dependence:** Modeling the dependence between variables is fundamental for understanding and analyzing their joint behavior. To achieve this, both parametric measures, such as the Pearson correlation coefficient, and non-parametric measures, such as rank-based correlations—Kendall's $\tau$ and Spearman's $\rho$—are employed. Non-parametric measures are particularly favored in the estimation of dependence for compound events because the marginal distributions of these data often deviate from normality. For instance, Kendall's $\tau$ is more appropriate when the joint distribution is non-Gaussian (Serinaldi,

2008). Spearman's rank correlation is based on the rankings of variable values, whereas Kendall's rank correlation assesses the concordance and discordance between pairs of observations (Czado, 2019). Both measures are valuable for evaluating the monotonic relationship between variables, indicating whether there is an increasing or decreasing trend in their joint values.

One limitation of non-parametric measures is that they do not reflect the full complexity of the dependence structure between variables. Specifically, these measures are not capable of reflecting tail dependence, that is, the strength of dependence in the

upper right or lower left quadrant of the bivariate distribution (Nelsen, 2006). This implies that non-parametric measures may not provide a complete assessment of dependence in these extreme regions of the joint distribution of variables (Poulin et al.,





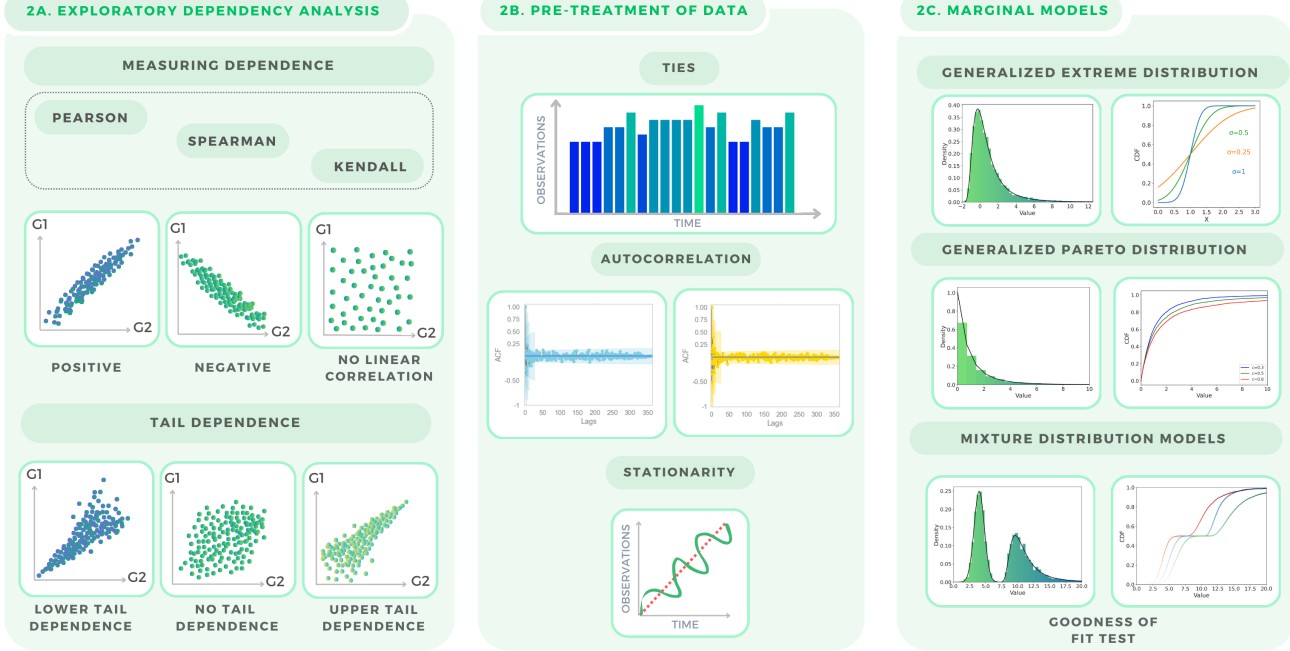

**Figure 2.** This figure expands on Step 2, Quantification of Compound Events, from Fig. 1, detailing the key processes involved in analyzing and preparing data. It includes 2A. Exploratory Dependency Analysis using Pearson, Spearman, and Kendall coefficients; 2B. Pre-Treatment of Data through handling ties, autocorrelation, and stationarity; and 2C. Marginal Models selection. Together, these elements provide a comprehensive approach for effectively quantifying and analyzing compound events.

2007). For this reason, the analysis of dependence in extreme compound events should be complemented using other methods, which will be discussed next.

**Graphical tools for analyzing dependence:** Several graphical methods are available to measure the dependence between
variables. Two of the most commonly used methods in the literature are chi-plots and K-plots. Chi-plots, first proposed by (Fisher and Switzer, 1985) and further elaborated by (Fisher and Switzer, 2001), are plots that rely solely on the data ranks and produce approximately horizontal diagrams under conditions of independence. These plots are useful for assessing the degree of association in the data, resembling control limits in an X-Bar chart.

Another graphical method inspired by chi-plots and proposed by (Genest and Boies, 2003), known as the Kendall plot or K-
plot, is an adaptation of the Q-Q plot concept. K-plots allow for the detection of dependence by analyzing the curvature present in the plot. Greater curvature indicates stronger association between the variables. K-plots offer the additional advantage of easily extending to multivariate cases and can adapt to time series contexts and situations where one of the variables is deterministic. K-plots retain several desirable features of the chi-plot, including their relationship with non-parametric tests of independence and their reliance on ranks.





**Tail dependence:** In the literature, various methods have been proposed to measure tail dependence from an extreme perspective, considering the probability of the joint occurrence of extremely small or large values. Frahm et al. (2005) examined several estimators for the tail dependence coefficient within a parametric, semi-parametric, and non-parametric framework. Schmidt and Stadtmuller (2006) provided non-parametric estimators to determine whether data exhibit tail dependence. Serinaldi et al. (2015) investigated the bias and uncertainty of multiple tail estimators proposed in the literature, highlighting their

limitations. They also found that the strong bias and associated uncertainty raise doubts about the reliability of most empirical results reported in the hydrological literature. Krupskii and Joe (2015) conducted a study on existing measures to evaluate the strength of tail dependence between pairs of variables. Lee et al. (2018) studied dependence measures arising from the extreme values literature to estimate the extremal coefficient.

Tail dependence measures can be used as additional information to general correlation measures, such as Kendall's $\tau$ and

Spearman's $\rho$, to facilitate the evaluation of a model that fits the data well in the tails (Krupskii and Joe, 2015). However, each measure has its applicability and usage limitations, so caution should be exercised when using them (Serinaldi et al., 2015). In the multivariate frequency analysis of extreme compound events, it is vital to consider the tail dependence structure of the variables of interest. If the selected copula fails to accurately capture this extreme dependence, there can be high uncertainty in the estimation of extreme values (Hangshing and Dabral, 2018). Therefore, conducting a thorough assessment

of tail dependence is crucial when selecting the most appropriate copula family for modeling extreme compound events. This approach ensures a proper model fit and reliable estimation of high-magnitude events in the context of floods. For more theoretical details, refer to (Czado, 2019), and for applied cases, see (Brunner et al., 2018; Hangshing and Dabral, 2018; Brunner et al., 2019).

### 3.2.2  Pre-treatment of data

**Ties:** Ties refer to the presence of identical or repeated values in a dataset. In the context of copula models, ties in the margins significantly impact rank-based inference. These ties not only reduce the sample size and efficiency of statistical procedures but can also alter the underlying dependence structure through selection bias (Genest et al., 2011). It is crucial to adequately consider ties, as their presence can definitively affect the fitting procedure (De Michele et al., 2013). Furthermore, ties can have a non-negligible impact on rank-based copula inference, as the performance of popular goodness-of-fit tests for copulas cannot

be guaranteed (Pappadà et al., 2017). Ties can negatively affect statistical data analysis and, worse, can render multivariate analysis questionable (Salvadori et al., 2014).

Various methods are discussed in the literature to address the issue of ties. Four proposed solutions include: (i) stopping the analysis, (ii) deleting observations with ties, (iii) using average ranks for ties, or (iv) breaking ties at random (Bücher and Kojadinovic, 2014). While these methods may seem intuitively straightforward to implement, it is important to recognize

that breaking ties randomly or using average ranks can introduce estimation biases (Li et al., 2020). Specifically, deleting observations with ties reduces the sample size, which can remove a significant portion of the data's spectrum and the information it contains. Furthermore, using average ranks from a large number of randomizations can be an insidious and potentially





misleading practice (Genest et al., 2011). Randomly breaking ties may not effectively identify the "true" model, particularly when the data is already discretized (Pappadà et al., 2017).

Due to the challenges posed by ties, other alternatives have been proposed in the literature. For example, Salvadori and De Michele (2006) and Salvadori et al. (2007) introduce methods that eliminate the lowest values of the variables to remove most ties in the data. Vandenberghe et al. (2010) propose a more holistic approach by introducing random noise into the observations. Kojadinovic (2017) adapted some existing statistical tests (tests of exchangeability, radial symmetry, extreme value dependence, and goodness-of-fit) to provide meaningful results in the presence of ties. Li et al. (2020) propose an

estimation method that treats the ranks of tied data as being interval-censored and maximizes a pseudo-likelihood based on interval-censored pseudo-observations. Considering the different proposals in the literature, it is clear that the solution to the problem of ties varies depending on the data and study variables. Therefore, there is no single solution, and this remains a topic that requires further research.

**Autocorrelation:** Autocorrelation refers to the degree of similarity or correlation between a time series and a lagged

version of itself at successive intervals. This can have a negative impact and pose several problems in the efficiency of copulas. Autocorrelation implies that observations are not independent of each other, which contradicts the basic assumption of independence between variables. According to Cong and Brady (2012), the presence of autocorrelation increases the variances of residuals and estimated coefficients, reducing the model's efficiency. To evaluate the serial dependence structure, one can apply the empirical autocorrelation function (ACF), which is a graphical method that shows the correlation between a data

series and lagged versions of itself. Another option is to use climacograms (Dimitriadis and Koutsoyiannis, 2015), which are diagrams where the variation of the time-averaged process over a range of time scales is plotted against the time scale. Additionally, other methods, such as the Ljung and Box (1978) test, can be applied to assess the significance of autocorrelation at multiple lags.

To manage the problems of autocorrelation, various techniques and models can be applied, such as moving average, autoregressive,

or autoregressive moving average models. The goal of these methods is to eliminate autocorrelation in the series. Additionally, if there is heteroscedasticity in the time series, Generalized AutoRegressive Conditional Heteroskedasticity (GARCH) models need to be applied. A GARCH model indicates second-order dependence in the time series, meaning that the conditional variability depends on the past history of the time series (Zhang and Singh, 2019). For more theoretical details, refer to (Zhang and Singh, 2019), and for practical applications, see (Cong and Brady, 2012; Serinaldi, 2016; Tootoonchi et al., 2022).

**Stationarity:** The non-stationarity of extreme compound events can result from various factors, including fluctuations in environmental variables, changes in climate patterns, and modifications in the interactions between variables. These changes can have significant impacts on various sectors, such as water management, agriculture, urban planning, and public safety. Understanding the non-stationarity of extreme compound events is crucial for adapting to and mitigating the associated risks. A detailed analysis of historical data and the use of advanced climate and statistical models are required to assess and project

non-stationarity in extreme compound events.

Climate variability can bias the estimation of extremes by partially invalidating the stationary assumption (Urrea Méndez and del Jesus, 2023). Several studies involve the non-stationarity of extreme climatic events (Mínguez et al., 2010; Solari and





Losada, 2011; Salas and Obeysekera, 2014; Sarhadi et al., 2016; Urrea Méndez and del Jesus, 2023), highlighting the scientific community's interest in this topic. This interest has driven significant progress in the research of non-stationary extreme values

over the past decades, providing experts in the field with greater capability to accurately describe both natural climate variability and the lack of seasonality.

The literature proposes various alternatives for combining multivariate analysis and non-stationarity. Examples include (Jiang et al., 2015), who proposed a time-varying copula model to investigate how reservoirs have altered the flow dependency structure in different locations along the Hanjiang River; (Lucio et al., 2020), who proposed a climate emulator considering

local wave climate variables (4-dimensional compound events), autocorrelated (on a daily time scale) and non-stationary (intra-annual variability); and (Jehanzaib et al., 2021), who evaluated the impact of non-stationary characteristics of meteorological drought on bivariate frequency analysis.

### 3.2.3   Marginal Models

The modeling of extreme values in marginal distributions is commonly approached through two main methods: using the

Generalized Extreme Value (GEV) distribution to capture block maxima over time, and the Generalized Pareto Distribution (GPD) within a peak-over-threshold framework (Coles, 2001). However, the literature recognizes that these traditional approaches do not always fit adequately when dealing with compound events, particularly in synchronous datasets that combine peak-over-threshold events across different stations (Brunner et al., 2019). In such cases, it is necessary to adapt or select methodologies that better reflect the complex interactions and dependencies among the involved variables (Davison and Huser, 2015). For

instance, while the GEV distribution has proven suitable for modeling extreme events at individual stations, its application in the context of compound events requires careful validation using statistical tests like the Anderson-Darling test to ensure that the model accurately captures extreme dependencies (Stephens, 1974).

Moreover, when combining maximum and non-maximum events within the same dataset, mixed models emerge as an alternative that may offer greater flexibility in fitting the distribution, by separating the behaviors of extreme values from more

common events (Scarrott, 2012). This is important to ensure that both maximum and non-maximum events are appropriately modeled in the analysis of compound events. However, much remains to be investigated in this area, and the best practice may be to experiment with different methodologies and apply a variety of statistical tests to determine the most suitable option for each specific case. Key tests include the Akaike Information Criterion (AIC), the Anderson-Darling test for assessing distribution fit, and the Kolmogorov-Smirnov goodness-of-fit test, which is useful for comparing the empirical distribution of

data with the proposed theoretical distribution (Genest and Favre, 2007; Czado, 2019).

### 3.3   Multivariate dependence structures

In this chapter, the steps involved in modeling the dependence between two or more variables using copulas are discussed. The methodology described integrates various approaches proposed in the literature (Genest et al., 2009; Brunner et al., 2016; Tootoonchi et al., 2022), which present the most common methods for selecting the appropriate copula model. Each step

also incorporates the concepts developed in previous chapters, making the described procedure adaptable to the modeling




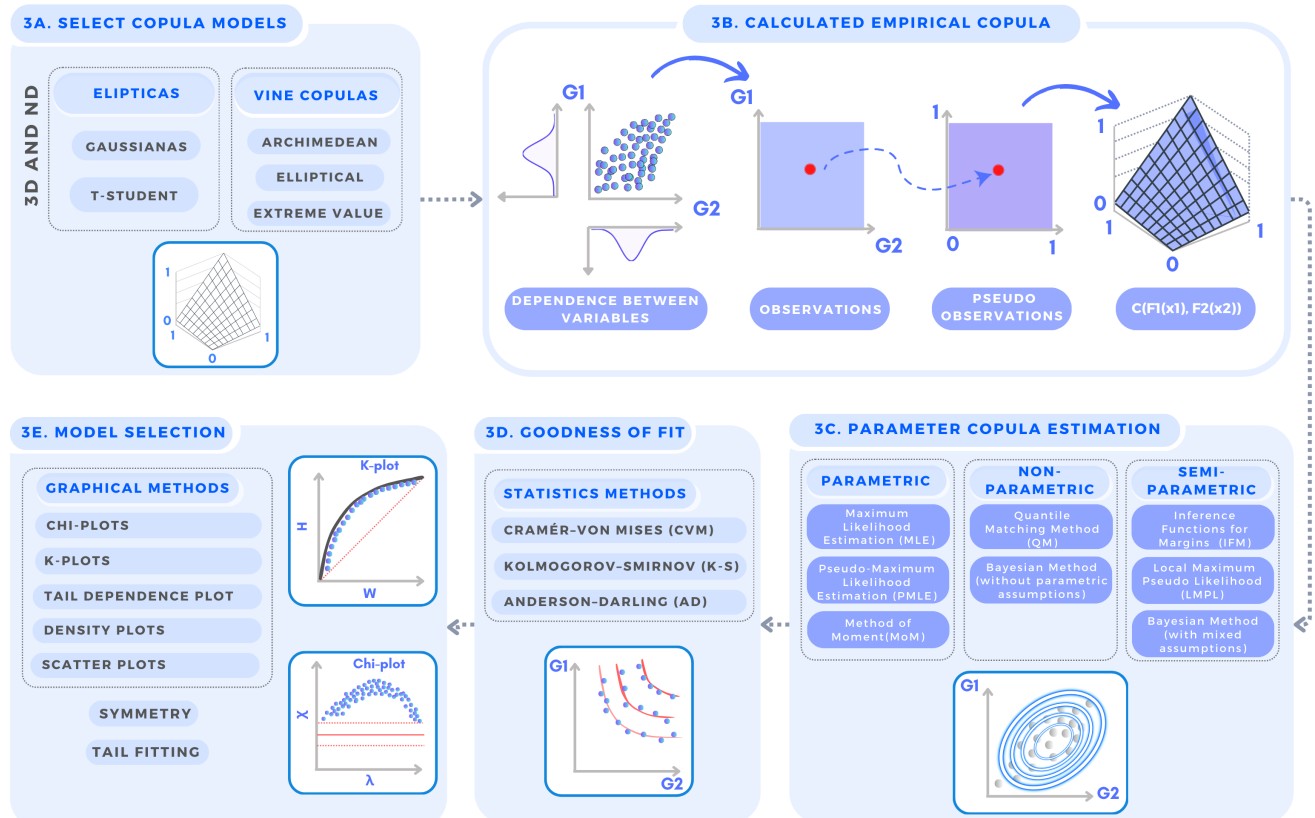

**Figure 3.** This figure provides an expanded view of Step 3 from Fig. 1, detailing the comprehensive framework for copula model selection and fitting. In step 3A, appropriate copula models (e.g., elliptical, vine) are selected based on data dimensionality. Step 3B involves calculating the empirical copula using pseudo-observations to capture the joint dependency structure. In 3C, copula parameters are estimated using parametric, non-parametric, and semi-parametric methods. Step 3D assesses the goodness of fit through tests like Cramér-von Mises and Kolmogorov-Smirnov. Finally, in 3E, model selection is refined using graphical methods, such as chi-plots and K-plots, to ensure accurate representation of dependencies and tail behaviors.

of compound events. Figure 3 provides a more detailed description of Step 3 from Fig. 1, illustrating the comprehensive framework for copula model selection and fitting.

- 3A. Selection of copula families: The selection of copula families for modeling compound events should be done according to the considerations in the chapter Evaluation of copula families for modeling dependencies in compound events.

- 3B. Calculate empirical copula: To capture the dependence structure (joint probability) between multiple sets of continuous random variables, such as $X, Y, ..., W$, it is common practice to use the empirical copula (Genest and Favre, 2007), which



characterizes the joint rank dependency between variables. Given the unknown nature of the marginal distributions, it becomes necessary to transform the original variables into the standard uniform space, as indicated by Eq. (1):

$$\hat{F}(x) := \frac{\mathrm{R_i}(x)}{n+1} \tag{1}$$

Where $\mathrm{R_i}$ denotes the rank among observations for a given variable, and $n$ is the number of observations. By applying the transformation described in Eq. (1), the variables $X, Y, ..., W$ are transformed into pseudo-observations $U_X, U_Y, ..., U_W$ in the range of a uniformly distributed $[0, 1]$.

- 3C. Estimation of copula parameter: The methods for copula parameter estimation are grouped into *parametric*, *non-*
*parametric*, and *semi-parametric* (Choroś et al., 2010). Their estimation is well explained in the literature, and for a deeper understanding of the various methods, refer to (Nelsen, 2006; Choroś et al., 2010; Weiß, 2011; Chen and Guo, 2019; Zhang and Singh, 2019).

- 3D. Goodness of fit: The next step involves evaluating the goodness of fit for a given copula. One approach is to generate random samples from the fitted copula family and compare the generated samples with the empirical copula data using
graphical methods (Genest and Favre, 2007; Brunner et al., 2016). As graphical methods provide an approximate measure of fit, it is necessary to apply goodness-of-fit tests such as Cramér-von Mises ($S_n$) with a bootstrap procedure (Wang and Wells, 2000; Genest and Favre, 2007; Genest et al., 2009). This test evaluates the null hypothesis that the empirical copula comes from a specific copula; if the null hypothesis is rejected, the empirical copula does not follow the distribution of the specified copula.

- 3E. Model selection: When applying goodness-of-fit tests, it is possible that several copulas may not be rejected. Subsequent selection criteria can include the Akaike Information Criterion (AIC) and, in the context of compound events, tail dependence tests. Additionally, performing a graphical analysis of the data, such as chi-plots or K-plots, can be valuable in evaluating whether tail dependence is present and assessing the symmetry of the data distribution. Symmetry analysis is particularly important because many copulas assume symmetric dependence structures, and deviations from
symmetry could indicate the need for alternative models. For vine copulas, a similar procedure is followed. A more detailed discussion of this topic is provided in (Brechmann et al., 2012; Czado, 2019).

## 3.4 Hazard scenarios

Once the key relationships between compound events influencing hazards and threats have been identified and analyzed, it is crucial to link them with the potential impacts they may cause. In the study of extreme compound events, various hazard
scenarios are employed to quantify their joint probability and magnitude. The most studied scenarios in the literature are the AND and OR scenarios (Salvadori and De Michele, 2004; Salvadori, 2004; De Michele et al., 2007; Salvadori et al.,




2011, 2016). These scenarios examine the possible combinations and effects of interactions between drivers when they reach extreme values, exceeding defined critical thresholds.

The AND scenario occurs when all drivers exceed a critical threshold and, therefore, occur simultaneously. This approach is
useful when analyzing the joint impact of two or more variables that must occur simultaneously to trigger an event of interest. The AND scenario can be appropriate for analyzing compound heat and drought conditions, which generate significant impacts on ecosystems when they occur simultaneously (Zscheischler et al., 2020). The OR scenario occurs when at least one of the drivers exceeds the critical threshold. This approach can be used for evaluating floods at the confluence of two rivers, where the occurrence of an extreme event in at least one or both rivers can generate a flood (Dodangeh et al., 2020).

The *Kendall* and *Kendall survival* scenarios are defined as a combination of the AND and OR hazard scenarios and are associated with occurrences that exceed the critical threshold or critical layer that defines the danger region. Additionally, they can be associated (solely) with any of the occurrences that are above the corresponding critical layer of level $t$ that identifies the dangerous region (Salvadori et al., 2016). This critical layer divides the sample space into a supercritical region and a non-critical region (Brunner et al., 2016). An application and analysis of the advantages and disadvantages of this approach
can be found in (Corbella and Stretch, 2012).

### 3.5 Multidimensional return period

#### 3.5.1 Bivariate return periods

Hazard scenarios are closely linked to multivariate return periods. Through an occurrence probability analysis, it is possible to calculate the events for each hazard scenario associated with a specific return level. In (Yue and Rasmussen, 2002; Salvadori
and De Michele, 2004; Salvadori, 2004; De Michele et al., 2007), a general definition of a return period applicable to both univariate and multivariate frameworks can be found, defining the return period of a 'hazardous' event as presented in Eq. (2):

$$T_D = \frac{\mu}{\Pr[X \in D]} \tag{2}$$

Where $D$ is a set of all hazardous values, $\mu$ is the mean time between two realizations of $X$, and the probability that a random variable $X$ is in the hazardous region $D$ is denoted as $\Pr[X \in D]$. According to [Brunner et al. (2016), the multivariate
return period can be defined from three approaches: The first approach refers to the calculation of the conditional return period and is generally used when one of the design variables is considered more important than the other (Salvadori et al., 2014). The second method uses joint probability distributions (copulas) to calculate JRPs. The third approach is based on the survival function or Kendall's distribution, although they are closely related to joint probability (Salvadori et al., 2016).

A more detailed explanation of the first approach can be found in (Brunner et al., 2016; Zhang and Singh, 2019). In the
second approach, hazard scenarios can be addressed using copulas. In the bivariate case, the AND and OR scenarios can be defined in terms of the JRP using equations Eq. (3) and Eq. (4), respectively.



$$T_{AND}(x,y) = \frac{\mu}{\Pr[X > x \, \text{and} \, Y > y]} = \frac{\mu}{1 - F_X(x) - F_Y(y) + F_{XY}(x,y)} = \frac{\mu}{1 - u - v + C(u,v)} \tag{3}$$

$$T_{OR}(x,y) = \frac{\mu}{\Pr[X > x \, \text{or} \, Y > y]} = \frac{\mu}{1 - F_{XY}(x,y)} = \frac{\mu}{1 - C(u,v)} \tag{4}$$

Where $\mu$ is the mean time between two realizations, $X$ and $Y$ represent conditional events, $x$ and $y$ are the critical thresholds,

$u$ and $v$ represent $F_X(X)$ and $F_Y(Y)$, which are the conditional events through the probability integral transform, and $C(u,v)$ is the selected copula model representing the joint distribution function $F_{XY}(x,y)$. These definitions can be extended to $d$ dimensions.

In the case of *Kendall* and *Kendall Survival*, they represent the intersection and union of infinite AND and OR hazard scenarios, respectively (Salvadori et al., 2016). The Kendall function defined in Eq. (5) for a multivariate case was introduced

by Salvadori and De Michele (2010) and represents the cumulative distribution function of the isolines (bivariate case) or hypersurface (more than two variables) of the copula.

$$K_c(t) = \Pr[W \le t] = \Pr[C(U,V) \le t] \tag{5}$$

Where $W$ represents the copula function $C(U,V)$, and $t$ is the critical probability level that divides the space into a supercritical region and a non-critical region. The Kendall return period is defined as the mean time between arrivals of critical

events at the probability level $t$ and is given in Eq. (6):

$$T_{K_c} = \frac{\mu}{1 - K_c(t)} \tag{6}$$

Since the Kendall function does not guarantee that all design variables are finite and bounded (Gräler et al., 2013), the Kendall's survival function was proposed by Salvadori et al. (2013). This function defines regions of 'safe' events that are bounded, ensuring that non-hazardous occurrences take bounded values of the analyzed variables (Salvadori et al., 2016).

Kendall's survival function is given by Eq. (7).

$$\bar{K}_c(t) = \Pr[S_{XY}(X,Y) \ge t] = \Pr[C(1-U,1-V) \ge t] \tag{7}$$

Where $S_{XY}$ is the survival function of $X$ and $Y$, and $t$ is the critical probability level. The return period of Kendall's survival uses the survival Kendall's distribution function and is defined in Eq. (8):

$$T_{\bar{K}_c} = \frac{\mu}{1 - \bar{K}_c(t)} \tag{8}$$





where the factor $1 - \bar{K}_c(t)$ yields the probability that a multivariate event will occur in the supercritical region (Brunner et al., 2016).

    The univariate and bivariate approaches differ in how they address the calculation of the return period. In the univariate approach, a unique solution for the design variables associated with $T$ is sought. In contrast, the bivariate approach is not limited to a unique solution but allows achieving JRPs through various combinations of the two involved random variables.

The JRP between two variables can be illustrated using contour lines (see Fig. 1 - step 5). These isolines represent a constant conditional return period and can be depicted for the OR, AND return periods, and Kendall definitions. Extending this concept to $d$-dimensional spaces poses a challenge, which we will address in the next chapter.

### 3.5.2   Higher dimensional return periods

The mathematical definitions of the bivariate return period are applicable to $d$-dimensional spaces (Brunner et al., 2016).
Contour lines representing the critical layer for a given return level become surfaces when the concept is extended to 3 dimensions. Beyond three dimensions, the representation of the critical layer becomes a hypersurface with a more mathematical than graphical definition.

    In the literature, several authors have extended the concept of the bivariate return period to multidimensional contexts (three-dimensional) using copulas and the Kendall measure. For example, Salvadori et al. (2011) describe how to approach this and
highlight the high computational cost involved. Other studies, such as that by Gräler et al. (2013), have used vine copulas and Kendall functions to jointly model the behavior of three hydrological variables. These studies share the approach of applying the return period concept in three dimensions but also face the mathematical complexity associated with describing the critical layer.

    To address the JRP for more than three variables, no formal definition has been established so far, and indeed, in our search,
we have not found studies focusing on this problem. Limiting analyses to only three variables may constrain and reduce the understanding of impacts generated by multidimensional or multiscale processes. The literature shows that increasing the number of variables brings complications in mathematical formulations (Grimaldi and Serinaldi, 2006). Selecting appropriate temporal and spatial scales for an event of interest is a challenge in itself, and this task becomes even more difficult in higher dimensions (Zscheischler et al., 2020).

To overcome this challenge, it is essential to consider alternative methods such as machine learning that allow for a simpler characterization of the critical layer so that it is possible to evaluate the joint occurrence of multidimensional/multiscale compound events in more than 3 dimensions and estimate the probability of their joint occurrence. In the next chapter, we will address this problem and propose a methodological framework for its solution.

### 3.5.3   Defining the critical hypersurface

The mathematical definition of the critical layer has been widely studied in two-dimensional spaces, in a few cases in three-dimensional spaces (Salvadori et al., 2011), and, at least at the time of this article's publication, we have not found a study that





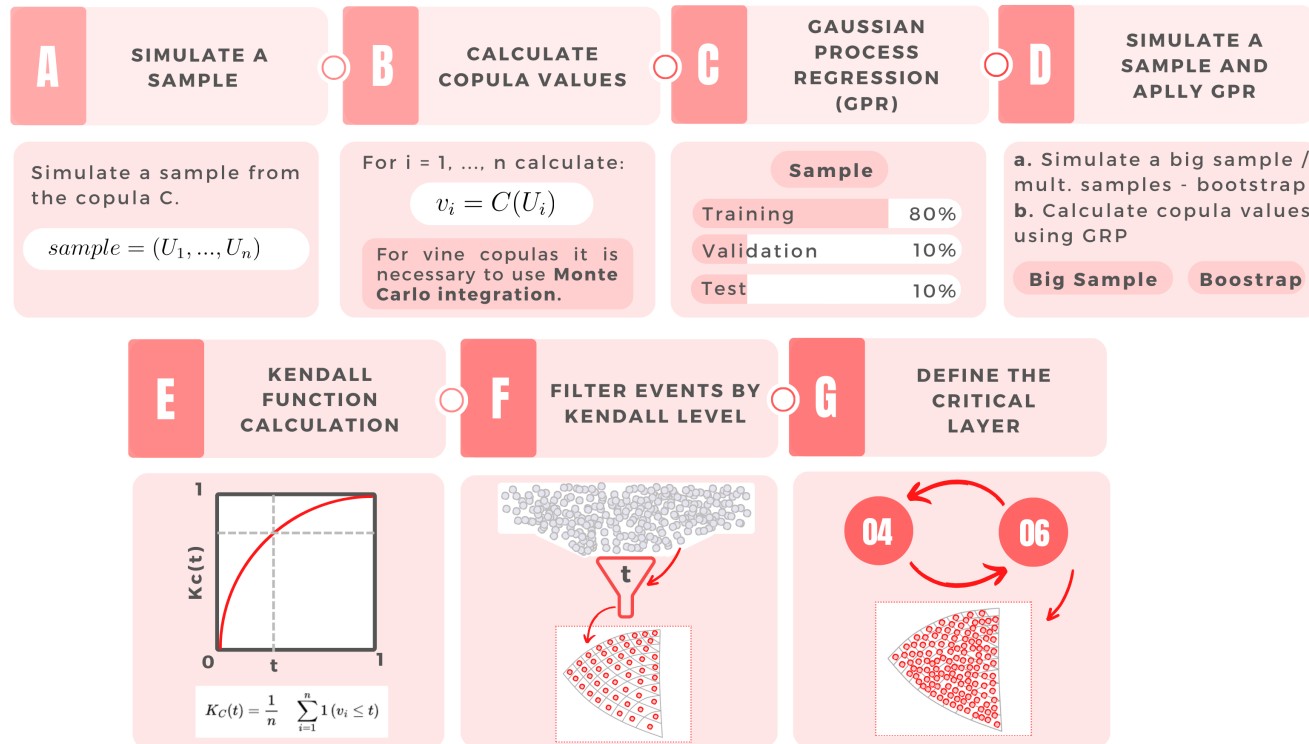

**Figure 4.** Methodological framework for defining the critical layer in multivariate spaces. The process begins with simulating a sample from the selected copula model (Step A). Each sample is then used to calculate the corresponding copula values (Step B). Given the computational complexity, Gaussian Process Regression (GPR) is employed to efficiently predict copula values for large datasets (Steps C and D). The Kendall function is calculated to determine the critical probability level $t$ associated with the return period (Step E). Events are filtered based on this critical probability level, defining the critical layer (Steps F and G). This process may be iterated to refine the critical layer, especially in high-dimensional spaces.

defines the critical layer in more than 3 dimensions. The critical layer in 2D is represented by an isoline, in 3D by a surface, and in higher dimensions by hypersurfaces that do not have a graphical representation but do have a mathematical one.

Generating the critical layer involves creating all possible combinations of events that make it up. These events can be generated for the hazard scenarios described in previous chapters and for a specific JRP. In this chapter, we will focus on the Kendall return period, as it involves an additional function compared to the other approaches. However, the proposed methodology can be easily adapted for the AND and OR scenarios. The proposed methodological framework is presented in Fig. 4 and consists of 7 steps. This figure provides a more detailed explanation of Step 5 from Fig. 1.

– 5A. Simulate a sample using the selected copula model, requiring an adequate sample size. According to Serinaldi (2013), at least 10,000 observations are needed to obtain reliable estimates of an isoline in the two-dimensional case. As



dimensionality increases, the sample size must also increase. The more variables included in the analysis, the larger the simulated sample size must be to adequately define the critical layer.

– 5B. Once the sample is generated, calculate the corresponding copula value for each set of values**. In the case of a vine copula, where a joint distribution function with a closed mathematical definition is not available, the Monte Carlo method is used to integrate the joint probability function and obtain the necessary copula values.

– 5C. Given that calculating the copula value for all simulated samples can be computationally expensive, an efficient approach based on the generic supervised learning method known as Gaussian Processes (GPR) (Rasmussen and Williams, 2006) is adopted for regression purposes. It is crucial to select an appropriate kernel and subsequently train and validate the GPR model.

– 5D. With the trained GPR model, the copula value can be calculated on the simulated samples at a very low computational cost. This optimization allows for the simulation of large sample sizes in a short time, making it relatively easy to implement bootstrap methods to calculate the critical Kendall probability level $t$, even in multidimensional spaces.

– 5E. Calculate the Kendall function from the simulated samples in step 4 and obtain the critical probability level $t$ associated with the selected return period. To perform this calculation, numerical methods may be required, as described in the work of Salvadori et al. (Salvadori et al., 2011).

– 5F. From the critical probability level $t$, filter the event combinations generated in step 4. These events constitute the critical layer.

– 5G. When filtering the events corresponding to a critical probability level $t$, a low number of events may be obtained. To increase the number of events that define the critical layer, steps 4 and 6 must be iterated.

With the described algorithm, it is possible to generate even millions of data points, from which the critical level $t$ can be filtered to obtain a better definition of the critical layer.

## 3.6 Compound design events

Based on the definition of the critical layer corresponding to a defined JRP, various alternatives for selecting design events can be found in the literature. Before describing these approaches, it is important to understand that the critical layer contains multiple events corresponding to the same JRP. This means that each combination of events contains the values that the drivers can take when the selected return period occurs. Due to this large number of possible events, (Salvadori et al., 2011) proposed two different approaches. The first approach involves selecting a single design event considered the most probable, taking into account the density of the multivariate distribution. The second approach involves selecting a set of events with the same return period, meaning events that lie within the critical layer. In this chapter, we expand the current methodological framework and propose various methods to extend the described approaches to $d$-dimensional spaces.



### 3.6.1 Most probable compound event

The simplest way to select the most probable design event among all possible options is to choose the point with the highest joint probability density. The procedure in two dimensions is straightforward: calculate the critical layer (isoline) corresponding to the selected return period. Then, for all pairs of variables $(u, v)$ that form this isoline, calculate the probability density. The pair of variables with the highest density is selected. This two-dimensional concept can be extended to $d$ dimensions using Eq. (9):

$$(u, v, ..., w) = \underset{C(u,v,...,w)=t}{\mathrm{argmax}} f_{XY...W}\left(F_X^{-1}(u), F_Y^{-1}(v), ..., F_W^{-1}(w)\right) \tag{9}$$

Where $f_{XY...W}$ represents the joint density function of the copula, $F_X^{-1}(u)$ represents the inverse of the fitted marginal distribution of variable $X$, evaluated at the variable transformed to the uniform space $(u)$. Applying Eq. (9) in more than two dimensions is challenging due to the high computational cost of defining the events contained in a $d$-dimensional critical layer, which becomes more complex as dimensionality increases. To select the set of most probable events, we propose two options: the first involves implementing computational optimization techniques, and the second involves increasing the resolution of the critical layer.

– Computational optimization techniques: To find the point with the highest probability density in the critical layer, computational optimization techniques can be employed. These techniques allow for systematic exploration of the search space to identify the optimal values that maximize the joint probability density function. A common approach is to use an optimizer that incorporates optimization algorithms to search for efficient solutions. In this case, the objective function is the function that defines the probability density of the copula. For the search to be efficient, it is important to specify the boundaries and constraints of the search space. The optimizer will iterate over different candidate solutions, evaluating the probability density of each set of events, iteratively exploring values to maximize the joint probability density in the critical layer. At the end of the optimization process, the optimal point that maximizes the probability density will be obtained, representing the most probable design event associated with the selected JRP.

– Refining the critical layer resolution: Another option for identifying the point with the highest probability density is to increase the resolution of the critical layer associated with the JRP under study. Computationally, this solution might intuitively be more costly due to the increase in event combinations it entails. However, by applying the methods for defining the critical layer described in Fig. 4, it is possible to identify a significant number of events over which Eq. (9) can be applied with relatively low computational cost. As the dimensionality increases, it is necessary to increase the number of event combinations to effectively define the critical layer and find the combination of events with the highest probability density.





### 3.6.2 Compound events ensemble

In many scenarios, choosing the event with the highest probability can result in a notable reduction of information obtained from a multivariate analysis (Gräler et al., 2013). To adequately address the variability of a subset of critical events, approaches such as the one presented by Volpi and Fiori have been proposed (Volpi and Fiori, 2012). In this chapter, we propose an approach that allows for obtaining a sample of possible design events from the critical layer and the copula's joint density function.

The importance of using multiple datasets instead of just the most probable events has been highlighted in previous works (Salvadori et al., 2011; Gräler et al., 2013). The critical layer contains all events $(u, v, ..., w)$ with the same JRP; however, not all have the same probability density. Using the copula's joint density function ($f_{XY...W}$), it is possible to sample a set of events, where combinations with higher density will be located towards the center and less probable ones at the edges of the critical layer.

To select events from the critical layer, we propose using the Metropolis-Hastings method. This approach is widely used to generate samples from a given probability distribution when direct sampling is not feasible. The main idea behind this method is to generate a sequence of candidate events based on a proposed probability distribution and then accept or reject these candidate events according to certain acceptance probabilities.

In this specific case, the Metropolis-Hastings method should be applied using the copula's joint density function ($f_{XY...W}$) as the target probability distribution. For each iteration of the algorithm, a set of candidate events within the critical layer should be generated, and the ratio between the probability density of the candidate events and the probability density of the current event should be calculated. Based on this ratio and an acceptance probability, it is decided whether to accept or reject the candidate events.

By repeating this iteration process multiple times, a sample of possible design events that represent the variability within the critical layer is obtained. This sample of events will be valuable for evaluating the uncertainty associated with the design of specific variables and will provide a more comprehensive view of extreme events within the multivariate context.

An initial limitation of this approach was longer execution times. However, we have significantly improved the computation times of the critical layer, which has considerably reduced the computational cost. According to Gräler et al. (2013), significantly more information is lost in three dimensions when selecting only one design event compared to the two-dimensional scenario. As dimensionality increases (more than three dimensions), it becomes imperative to analyze a set of events, which can be obtained through this analysis. This improvement is especially relevant in the event selection step, as it allows for a more complete and accurate representation of the variability within the critical layer.

## 4  Discusion

One of the main objectives of this study has been to complement the existing methodological framework for the analysis of compound extremes and JRPs. In particular, we have provided practical guidelines and detailed steps to specifically address events involving two or more hydrological drivers. While univariate analysis has been widely used in engineering to estimate





impacts, we have highlighted the need to move towards a more comprehensive approach that considers the interaction of multiple variables.

The definition of the critical layer and the JRP in the context of compound extreme events is of utmost importance for hydrological risk analysis and management. As mentioned in the introduction, compound events result from complex interactions between multiple variables across various temporal and spatial scales. These events can have significant impacts on the environment and society, underscoring the need to accurately estimate their magnitude and provide precise design variables.

The critical layer, as described, represents a set of events that share the same JRP. In other words, they are combinations

of hydrological variables that have a specific probability of occurring simultaneously with a determined frequency. The identification and characterization of the critical layer are essential for understanding and evaluating the impacts of compound extreme events, as well as for making informed decisions in risk planning and management.

The JRP, on the other hand, refers to the analysis of the simultaneous occurrence of multiple extreme events and how their joint probability of occurrence varies over time. It is a fundamental measure for hydrological risk estimation and provides

valuable information on the frequency and severity of compound events. Determining the JRP involves analyzing the dependence between the hydrological variables involved, which requires advanced statistical and mathematical approaches.

This study has proposed a methodological approach that integrates hydrological, statistical, mathematical, and machine learning concepts to address the challenges of JRP analysis in more than three variables. The definition of the critical layer or hypersurface associated with the JRP in n dimensions, using Kendall's function and a computationally efficient method, has

been one of the main objectives of this work.

By obtaining a sample of possible design events that represent the variability within the critical layer, we have been able to evaluate the uncertainty associated with the design of specific variables and provide a more comprehensive view of extreme events in a multivariate context. This approach has proven valuable in quantifying hydrological risks associated with compound events, allowing for a better understanding of their impact on society and the environment.

However, it is important to note that the analysis of compound events and the JRP in more than three dimensions remains a challenge and an area of ongoing research. Mathematical formulation and computational approaches become more complex as dimensionality increases. Despite this, the methodological advances presented in this study offer a solid foundation for future research and practical applications in the field of hydrology and risk analysis associated with compound extreme events.

## 5    Conclusions

Compound extreme events, such as floods, droughts, and wildfires, are complex hydrological phenomena caused by the interaction of multiple drivers across different temporal and spatial scales. Understanding and properly analyzing these events are crucial for planning and managing the associated risks.

In this article, we propose a methodological approach that combines hydrological, statistical, mathematical, and machine learning concepts to address the challenges of analyzing the JRP in more than three variables. The primary objective has been

to define the critical layer or hypersurface associated with the multivariate JRP in n dimensions, considering Kendall's function and using a computationally efficient method.

Through the use of copulas and the multivariate return period, we have been able to obtain a sample of possible design events that represent the variability within the critical layer. This sample of events is valuable for evaluating the uncertainty associated with the design of specific variables and provides a more comprehensive view of extreme events in a multivariate context.

The analysis of compound events and the JRP in more than three dimensions remains a challenge in hydrology and other disciplines. However, the methodological advances and efficient computational tools presented in this article offer a solid foundation for future research and practical applications in the field of hydrology and risk analysis associated with compound extreme events.

It is hoped that this methodological approach will contribute to a better understanding of compound events and their
impact on society and the environment, enabling more effective planning and management in the face of these complex and increasingly frequent phenomena due to climate change.

*Author contributions.* Manuel and Diego conceptualized the study and developed the theoretical framework. Diego also implemented the computational models and performed the data analysis. Dina and Diego designed and conducted the statistical tests. All authors contributed to the interpretation of results and the manuscript preparation.

*Competing interests.* The authors declare no competing interests.

*Disclaimer.* No potential conflict of interest was reported by the authors.

*Acknowledgements.* This work received financial support from the Government of Cantabria through the Fénix Programme and from Grant RTI2018-096449-B-I00, funded by MCIN/AEI/10.13039/501100011033 and by "ERDF A way of making Europe." Additionally, the author
used artificial intelligence tools to review the English and to generate part of Figure 1. These tools assisted in optimizing the clarity of the text and in the creation of the figure, without influencing the scientific content or analysis presented.





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
