# Peer review of "Return period of high-dimensional compound events. Part I: Conceptual framework - Supplemantary information"

_Hydrology and Earth System Sciences, 2024_

## Referee Comment (RC1)

**REVIEW REPORT**

**Journal:** HESS
**Paper:** hess-2024-334
**Title:** Return period of high-dimensional compound events. Part I: Conceptual framework
**Author(s):** Manuel Del Jesus, Diego Urrea Méndez, and Dina Vanessa Gomez Rave

**GENERAL COMMENT.**

For the reasons given below, I recommend a **REJECTION WITHOUT RESUBMISSION.**

This paper is useless, and introduces no novelties concerning the estimate of the RP of high-dimensional occurrences. The work does not contain any relevant advance: it simply suggests (in a non-technical and superficial way) to use well known algorithms to find the critical points of multidimensional functions.

Not to say about the "Conceptual Framework": it is already well known in hydrological/geophysical Literature since about 20 years. All the formulas presented (sometimes inexact from a mathematical point of view) do not show anything new, nor they represent any advance with respect to the present knowledge.

In addition, the fact that the occurrences considered are Compound Events does not emerge in any part of this work: only superficial comments and descriptions are presented, but the true core problem is nowhere investigated.

Furthermore, sort of indications for carrying out a multivariate analysis are sketched, but they are too generic, never discussed in details concerning the problems they are expected to solve, and most of all they are all already well known in Literature: this paper adds nothing to knowledge, all what is written has already been more precisely (and mathematically correctly) introduced in already published works, so what?

Finally, the mathematical notation is often wrong: e.g., the Authors confuse a function (say, $F$) with its value (e.g., $F(x, y)$), or even worse confuse $F(x)$ with $F(X)$, where the former is a real number, whereas the latter is a random variable.

Overall, the content of this paper can be summarized in a single paragraph, and recycled in the Introduction of the companion paper (Part II).

**SPECIFIC COMMENTS.**

**Line(s) 55–56**

**AUTHOR(s).** Vine copulas are a flexible class of dependence models consisting of bivariate building blocks.

**REFEREE.** The Authors do not mention the problems intrinsic to modeling via Vine copulas.

**Line(s) 58–59**

**AUTHOR(s).** For more theoretical details, please refer to (Sklar, 1959; Nelsen, 2006).

**REFEREE.** None of the references is pertinent to Vine copulas: more recent and relevant paper must be cited, starting from the basic one:

Aas, K., Czado, C., Frigessi, A., Bakken, H. (2009). Pair-copula constructions of multiple dependence. Insurance: Mathematics and Economics, 44(2), 182–198.

**Line(s) 149–151**

**AUTHOR(s).** For instance, Kendall's $\tau$ is more appropriate when the joint distribution is non-Gaussian (Serinaldi, 2008). Spearman's rank correlation is based on the rankings of variable values, whereas Kendall's rank correlation assesses the concordance and discordance between pairs of observations (Czado, 2019).

**REFEREE.** None of the references is pertinent. Much better references are the two books by Nelsen (2006)—a more theoretical one—and Salvadori et al. (2007)—a more practical one.

**Line(s) 159–ff.**

**AUTHOR(s).** Graphical tools for analyzing dependence...

**REFEREE.** For intellectual honesty, the Authors should warn the reader that the interpretation of graphical results always involves a degree of subjectivity, and should always be accompanied by objective formal tests.

**Line(s) 175–176**

AUTHOR(s). They also found that the strong bias and associated uncertainty raise doubts about the reliability of most empirical results reported in the hydrological literature.

REFEREE. See also Illustration 3.18 in Salvadori et al. (2007, p. 173), where numerical experiments were carried out both on empirical and simulated data.

**Line(s) 242**

AUTHOR(s). The literature proposes various alternatives for combining multivariate analysis and non-stationarity.

REFEREE. For intellectual honesty, the Authors should point out that, at present, non-stationarity is generally modeled by introducing a temporal dependence of the parameters of the marginals/copulas at play (e.g., by assuming linear and/or exponential changes of the parameters with time), but these remain mere mathematical exercises, not tested on empirical data.

**Line(s) 292–294**

AUTHOR(s). This test evaluates the null hypothesis that the empirical copula comes from a specific copula; if the null hypothesis is rejected, the empirical copula does not follow the distribution of the specified copula.

REFEREE. Statistically speaking, this sentence is not correct: Statistics can only offer guidance and suggestions, but never absolute truths. The words "copula does not" should be written as "copula may not".

**Line(s) 422–ff. (3.6 Compound design events)**

REFEREE. The Authors are quite confused about the difference between a density and a probability distribution function: they are not the same, and they have different meanings. For instance, in the cited paper by Salvadori et al. (HESS 2011), the two different strategies proposed were based either on a probabilistic base (the Component-wise Excess one) or on a likelihood/density base (the Most Likely one). The description and the explanation given by the Authors is a superficial and confusing one, especially for practitioners.

Here and elsewhere, use "most likely" instead of "most probable": in Probability Theory, a density induces a likelihood, not a probability (which, instead, is induced by a distribution function, i.e. the integral of a density). Line 427 refers to the Most Likely strategy outlined in the cited paper by Salvadori et al. (HESS 2011). Line 428 refers to the usage of "ensembles", as suggested in Salvadori et al. (HESS 2011).

Eq. (9) is a special case of Eq. (13) in Salvadori et al. (HESS 2011), using the density $f$ as a weight function $w$. Note that $f_{XY...W}$ is the joint density of the distribution $F_{XY...W}$, not of the copula $C_{XY...W}$.

For intellectual honesty, the Authors should make it clear, and warn the reader, that there is no guarantee that the maximum found by a numerical routine will be a global maximum, rather it is very likely that it will be a local maximum, and this will be more and more likely as the dimension of the problem increases.

**Line(s) 489–ff. (Discussion, Conclusions)**

REFEREE. This is not a Discussion, it is at most a replica of generic sentences already written in previous parts of the manuscript. Actually, in my opinion, in this paper there little to discuss about.

The Conclusions are a collection of statements that try to justify a paper with no content.

---

## Community Comment (CC2)

Dear Hafidha Khebizi,

Thank you very much for your comment and for sharing your interest in our work. We are pleased to know that you find our approach valuable for your research in the Algerian Sahara.

We would like to inform you that we have already published a second part of this study, which is currently under review and available as a preprint, titled *Return period of high-dimensional compound events. Part II: Analysis of spatially-variable precipitation*. You may consult it at the following link: https://hess.copernicus.org/preprints/hess-2024-335/.

In this work, we present a comprehensive framework for modeling compound precipitation events, including a practical application that addresses spatial dependence in precipitation across multiple sites and the estimation of multivariate return periods in five dimensions. We encourage you to review this second part, as it may provide useful insights into the questions you raise. Additionally, we are currently evaluating the methodology in combination with other multivariate approaches and continuous hydrological-hydraulic models, which we hope to share in future publications.

**Question 1 and 2**

To specifically address the questions, you posed—namely, **(1)** *how to integrate the floods in the province of Valencia into a risk scenario*, and **(2)** *how to incorporate the cold drop into the analysis of multivariate return periods according to our six-step approach*—we have developed a hypothetical case study that examines these issues and offers insights into both inquiries.

The flood event in the province of Valencia in October 2024 can be analyzed using the proposed methodological framework, structured into six key steps. In the ***initial diagnostic*** phase, this event can be classified as a spatially and temporally compound event. The main modulators were the Mediterranean Sea surface temperature and the presence of an isolated cut-off low, known in Spanish as "Depresión Aislada en Niveles Altos" (DANA). The interaction between these factors created the necessary conditions for extraordinary rainfall exceeding 200 mm in 24 hours, thereby amplifying extreme surface runoff. In terms of temporal resolution, sub-daily data are particularly relevant to capture the intensity of the rainfall, while spatially, it is crucial to include both urban areas and the affected river basin in the analysis.

Subsequently, an ***exploratory dependence analysis*** could be conducted using coefficients such as Kendall or Spearman to quantify the relationships among the key variables involved in the event, such as the elevated Mediterranean Sea surface temperature and the influence of the DANA (reflected in parameters like rainfall, wind, and atmospheric pressure). This analysis would, on the one hand, allow for the assessment of dependencies among multiple variables to identify the joint return period of the conditions that triggered the event. On the other hand, it could incorporate the factors driving the flooding, for example by considering precipitation dependencies measured at different locations. This approach would be especially useful for examining how the intensity and spatial distribution of rainfall contributed to the floods in Valencia, providing a more comprehensive understanding of the interactions that led to this extreme event.

The next step would involve ***identifying the most suitable multivariate structure*** to represent the behavior of the variables. To achieve this, it would be crucial to consider metrics such as upper- and lower-tail dependencies, along with other tools that ensure an excellent goodness of fit. This approach guarantees that the selected model accurately captures the relationships between variables, providing a robust and realistic representation of the phenomenon under analysis.

The fourth step would consist of calculating a variety of ***hazard scenarios*** using "AND" and "OR" approaches, complemented by analyses based on "KENDALL" and "KENDALL SURVIVAL" scenarios. These proposed scenarios would offer a more detailed understanding of the interactions among the main variables. Based on this foundation, for instance, one could explore an "AND"-type scenario in which elevated Mediterranean temperatures and the presence of a DANA (represented by multiple variables such as rainfall, wind, and atmospheric pressure) are necessary conditions for generating the compound event. Moreover, these approaches could be integrated into a spatial analysis, estimating the joint return period of rainfall across different locations. This would help identify the multivariate return period of the rainfall produced by the DANA, taking its spatial dependency into account.

Once the compound event and its genesis conditions are defined, it becomes important to distinguish it from its impacts, especially the potential flooding it could cause. To this end, a broader, multidimensional, and multivariable scenario could be evaluated—one that not only includes the initial meteorological variables but also those that capture the hydrological response and the formation of the flood itself, such as peak flow, total accumulated water volume, and event duration. Integrating these approaches with a spatial analysis that estimates the joint return period of rainfall at various locations not only provides information about the multivariate frequency of the conditions that triggered the event, but also helps anticipate how the spatial distribution of water contributes to the formation and extent of flooding.

Finally, by applying the framework proposed in Figure 4, we could ***define the critical layer*** for each of the proposed scenarios. From this critical layer, it would be possible to identify the compound event with the greatest probability density or to conduct a comprehensive analysis of joint events for the studied return period. For the scenario that considers elevated Mediterranean temperatures and the presence of a DANA, we could examine the period or frequency at which such a large-scale combined event might occur. If we consider the spatial dependency of rainfall, it would be possible to extract spatial rainfall events associated with the event's return period and, using rainfall-runoff models, simulate the basin's hydrological response and the resultant flooding. Finally, by considering a multidimensional and multivariable scenario, we could evaluate the joint behavior of all the described variables to assess compound events such as those that occurred on October 29 and 30 in terms of risk.

This framework would provide a broader range of potential events, useful not only for establishing response and mitigation protocols but also for guiding the redesign of hydraulic infrastructure and the development of early warning systems to address similar scenarios in the future. This comprehensive approach combines scientific understanding of the phenomenon with practical applications, ultimately improving resilience against extreme hydrometeorological risks.

*Question 3*

Regarding your third question—how to select the variables and temporal/spatial scales to define compound extremes and present a risk scenario for a cold drop (DANA)—it is important to consider both meteorological and hydrological factors, as well as the appropriate resolution and coverage in time and space. In the following explanation, we outline the key considerations that ensure a comprehensive and reliable assessment of such extreme events.

To define compound extremes associated with the spatial evolution of a "cold drop"—a colloquial term historically used in Spain to describe episodes of intense rainfall, but which today is generally associated with the meteorological phenomenon known as an isolated cut-off low (DANA)—it is necessary to draw on variables that capture both its meteorological and hydrological characteristics. Meteorologically, it is common to include factors such as precipitation intensity and duration, Mediterranean Sea surface temperature, atmospheric pressure, as well as wind direction and speed; all of these influence the formation and persistence of the phenomenon. From a hydrological perspective, variables such as peak river discharge, total accumulated rainfall volume, surface runoff, and soil drainage capacity are fundamental for understanding the event's territorial impact and the potential flood risk.

The selection of temporal and spatial scales must be adapted to the dynamics of the DANA and the objectives of the risk analysis. Temporally, sub-daily resolutions (e.g., hourly) allow for capturing the event's peak intensity and rapid evolution, which is essential for identifying precipitation peaks or critical moments in flood formation. At the same time, supplementing these data with daily or weekly information makes it possible to assess longer-term trends and recurrences over extended return periods. Furthermore, analyzing long-term time series encompassing multiple DANA events over several decades can provide a more robust statistical foundation, improving the understanding of long-term variability and the reliability of return-period estimates across different temporal scales.

From a spatial standpoint, it is important to consider various levels of detail: local scales to understand immediate hydrological responses in urban areas or small watersheds, and regional scales to evaluate the extent, displacement, and interaction of the DANA with surrounding atmospheric conditions, as well as its impact on multiple river basins.

In short, carefully selecting variables, as well as temporal and spatial scales, contributes to more accurate risk scenarios. These scenarios integrate not only the identification of the extreme event and its atmospheric origin, but also the spatial distribution of its effects, the temporal dimension of its evolution, and the inclusion of extensive historical records. Altogether, this provides an effective tool for designing mitigation strategies and improving risk management in the face of extreme hydrometeorological phenomena.

Please, do not hesitate to contact us if you need any additional clarification

Best regards

---

## Author Comment (AC1)

**Rebuttal of Review 1**

Dear reviewer,

Below you will find our comments and responses to your review.

**General Comment**

**This paper is useless, and introduces no novelties concerning the estimate of the RP of high-dimensional occurrences. The work does not contain any relevant advance: it simply suggests (in a non-technical and superficial way) to use well known algorithms to find the critical points of multidimensional functions.**

We do not agree with your valuation of our contribution as *useless*. This paper is the first part of a two parts paper, and we consider this distinction crucial to understand the role that our work intends to play. In our experience, multidimensional statistics, and specifically extremes, is a complicated subject where, if it is true that a lot of literature exists, more often than not methodological details are missing. Our aim with this paper was twofold. First, we wanted to make sure that our methodology was fully reproducible, and for that, we wanted to make sure that all the relevant points were covered. Due to the fragmentation of much of the research on the topic, we considered to present an extended methodology that anyone could follow. The second aim, once we realized the extension of such a methodology, was to try to be as inclusive as possible and carry a literature review of the different options that exist. In this way we provide the reader with a more clear picture of the choices made in the paper, without giving the impression that our way to proceed forward was the only one.

Even considering both parts, it may seem that this paper includes more additional details than needed, and that is because there is a third paper, were we deal with other different variables, were the selection of events and the treatment of the distribution are quite different. Thus, including all the required information into this paper to make references to it from all our posterior works. In this regard, we were discouraged to submit the third paper as a third part, but we can make it available to the reviewers should you consider to evaluate if our argumentation is more understandable with all the judgement elements at hand.

In response to concerns regarding the novelty and relevance of our work, we find it essential to make clear how our research not only integrates existing literature but directly addresses one of the key limitations identified in the field: the estimation of multivariate return periods in high dimensions and the definition of the associated critical layer (or hypersurface). As noted by G. Salvadori, De Michele, and Durante (2011), as well as in recent studies (Gräler et al. (2013); Brunner, Seibert, and Favre (2016), Xu, Wang, and Bin (2023); Brunner (2023)), there is a consensus within the scientific that the mathematical and computational challenges of extending and aplying existing techniques to higher-dimensional spaces (n>3) remain underexplored. Our paper advances in this area through:

1. *Integration of existing methodological approaches for joint return periods (JRPs)*: We present a comprehensive approach that organizes and synthesizes existing methodologies, offering practical guidelines and structured steps to enhance reproducibility and applicability. Given the complexity and fragmentation of methods in the literature, this work aims to provide a well-documented and cohesive framework that integrates statistical and hydrological techniques, ensuring broader accessibility for researchers and practitioners.

2. *Definition of the multidimensional critical layer*: We propose a novel approach to define the critical hypersurface associated with compound events in high-dimensional spaces. This approach integrates advanced statistical techniques with Gaussian Process Regression (GPR), thus optimizing the computation of the critical layer and significantly reducing the associated computational cost.

3. *Reduction of computational cost*: Unlike traditional approaches that rely on intensive simulations, our methodology employs a combination of Monte Carlo simulation and GPR-based regression models. This enables the efficient computation of copula values in large datasets, making high-dimensional analysis feasible without compromising accuracy.

4. *Generalizable and applicable methodology*: Although our methodology is initially presented for analyzing the spatial dependence of rainfall regimes, its flexible and adaptable design allows for its application to different types of compound events, including multivariate, preconditioned, temporally compounding, and spatially compounding events, as classified by Zscheischler et al. (2020). Thanks to its adaptability, our approach is applicable to a broad range of environmental and climate-related challenges.

5. *Opening avenues for future research*: As detailed in our paper, once the critical layer in high-dimensional spaces is established, it opens the door for more detailed studies on the selection of design events and uncertainty analysis. We thus consider that our contribution is neither superficial nor a mere application of well-known methods. Rather, it constitutes a methodological advancement that tackles a well-documented challenge in the scientific literature while providing practical tools for implementation.

We would also like to emphasize that this work constitutes the first part of a broader investigation, with the second and third parts presenting specific results from its application to real-world case studies. This integrated and efficient approach is not only novel but also establishes a solid foundation for future research and practical applications in the analysis of high-dimensional compound events.

**Not to say about the "Conceptual Framework": it is already well known in hydrological/geophysical Literature since about 20 years. All the formulas presented (sometimes inexact from a mathematical point of view) do not show anything new, nor they represent any advance with respect to the present knowledge.**

We acknowledge that the "Conceptual Framework" is well known in the hydrological and geophysical literature, having provided a solid theoretical basis for the past two decades. However, our contribution does not lie in *redefining* this framework,

but rather in its integration into a novel methodological approach designed to address specific challenges in high-dimensional spaces. By combining this theoretical foundation with advanced regression techniques (such as Gaussian Process Regression), efficient computational simulations, and an optimized approach for defining critical layers, we provide practical solutions that are not directly available in the existing literature.

Moreover, reproducibility is a cornerstone of scientific progress, and incorporating established formulas ensures that our work remains transparent, accessible and verifiable by the broader research community. However, what sets our contribution apart is not merely the use of these concepts but their targeted application and the significant advancements in computational efficiency. These innovations provide deeper insights into the analysis of multivariate return periods in hydrology, and represent an ongoing effort to overcome persistent limitations that have yet to be fully addressed in the field. While we build upon two decades of research, our work strives to go beyond a mere repetition of existing methods. Rather, it seeks to integrate, adapt, and optimize established approaches to to enhance the reliability of multivariate return period estimation.

**In addition, the fact that the occurrences considered are Compound Events does not emerge in any part of this work: only superficial comments and descriptions are presented, but the true core problem is nowhere investigated. Furthermore, sort of indications for carrying out a multivariate analysis are sketched, but they are too generic, never discussed in details concerning the problems they are expected to solve, and most of all they are all already well known in Literature: this paper adds nothing to knowledge, all what is written has already been more precisely (and mathematically correctly) introduced in already published works, so what?**

We appreciate the reviewer's comments and understand the concern about the "true core problem," which, however, seems somewhat ambiguous to us, as it is not entirely clear which specific aspect the reviewer is referring to. From our perspective, one of the many fundamental problems in this field—widely discussed in the literature— is the definition of the multidimensional critical layer and its effective application to hydrological problems involving compound events. Our research is grounded in the pivotal contributions of G. Salvadori, De Michele, and Durante (2011), Gräler et al. (2013), Brunner, Seibert, and Favre (2016), Xu, Wang, and Bin (2023), and Brunner (2023), but seeks to extend this methodological framework by addressing several key limitations:

1. *From theory to a reproducible and practical methodology:* While previous studies have emphasized the theoretical importance of multivariate return periods and critical surfaces, they often lack practical guidelines on how to define these layers in high-dimensional spaces. To the best of our knowledge, no study has developed and applied a general methodology, including vine copulas, to derive multivariate return periods beyond three-dimensions, highlighting the challenges in this field. These challenges include high computational costs associated with managing complex dependence structures and the lack of a clear methodological framework to efficiently address these issues. Our work bridges this gap by providing a detailed and reproducible procedure to identify the critical layer, integrating statistical models and computational optimizations that make its application feasible in higher-dimensional settings.

2. *Applications in hydrology and coastal systems:* While the general framework for compound events is well established, our contribution lies in its adaptation to hydrological problems with high-dimensional dependencies, such as rainfall regimes and their spatial distribution. Moreover, the proposed methodological framework may be extendable to the study of compound events in coastal and estuarine environments, where interactions between river discharge, storm surges, and extreme winds can have significant impacts on coastal dynamics and infrastructure.

3. *Challenges in high-dimensional modeling:* As noted by Brunner (2023), modeling dependence in multivariate environments is feasible in low dimensions (e.g., bivariate cases) but becomes increasingly complex and computationally demanding as the number of interdependent variables increases. Identifying suitable dependence structures in high dimensions is not always straightforward and requires more flexible approaches to simultaneously account for temporal, spatial, and variable dependencies. In fact, these aspects have been recognized in the literature as *outstanding challenges and future research directions* (Brunner 2023), further reinforcing the relevance of our work in addressing these issues through computational optimizations, including efficient Monte Carlo simulations and regression-based models, enabling scalable and accurate analyses within the critical layer framework.

4. *Compound floods with multiple drivers:* Xu, Wang, and Bin (2023) highlight that, in the context of climate change and urbanization, the interaction between extreme precipitation, storm surges, and rising sea levels will continue to intensify, making the study of compound events in coastal environments increasingly relevant. However, most studies have focused on two-dimensional scenarios (e.g., rainfall-storm surge or runoff-storm surge), while three-dimensional scenarios still present *methodological and computational challenges.* Our work contributes to close this gap by developing methodologies for identifying critical layers in *high dimensions*, allowing for a more realistic approach and improving the predictive capabilities of hydrological models .

This article represents the first part of a three-stage study. In this work, we establish the general framework and methodology for identifying the critical layer in high-dimensional compound events. The second part of this study, currently under review, focuses on applying the methodology to spatially and temporally compounding events, providing detailed numerical validations and case studies. Additionally, there is a third part of the study, which, although not submitted to this journal, applies the same methodological framework to multivariate compound events in estuarine environments, considering oceanic, hydrological and meteorological variables to analyze interactions between different extreme phenomena affecting these ecosystems.

From this perspective, our work complements and extends previous studies rather than replicating them. By making existing knowledge more accessible and applicable to hydrological, and more generally, environmental problems, we aim to contribute to the ongoing scientific effort to better understand and manage compound events in complex environments.

**Finally, the mathematical notation is often wrong: e.g., the Authors confuse a function (say, $F$) with its value (e.g., $F(x, y)$), or even worse confuse $F(x)$ with $F(X)$, where the former is a real number, whereas the latter is a random variable.**

We will revise the mathematical notation. It is true that none of the authors is a mathematician by training, and that some expressions were not rigorous. Luckily, these imprecisions pertain to the realm of wording and have not introduced any conceptual error into the methodological procedure itself.

**Overall, the content of this paper can be summarized in a single paragraph, and recycled in the Introduction of the companion paper (Part II).**

Again, we do not agree with you. Although we perfectly understand that highly-trained researchers in multivariate statistics may find some of the contents unnecessary and summarizable in a single paragraph, in our view, most hydrologists require more details and explanations to approach such a complex topic. For instance, the review for Part II mentions that we should not include approaches that are wrong just because practitioners use them. In our opinion, we researchers have a deontological obligation to convey our knowledge in the most comprehensible way possible. If there are professionals using these techniques wrongly, it may be because of their intellectual limitations, but the hypothesis that we favor is the one in which the problem is that nobody cared to build a better bridge for them to transit from lousy practice to rigorous application.

We stress the importance of providing an extensive and detailed presentation of the methodology to address the knowledge gap that still exists in the practical application of multivariate statistics in hydrology. While many of the concepts discussed may be familiar to experts in multivariate statistics, our goal is to present them in a clear and applicable manner for a broader audience, including readers and professionals in hydrology and coastal studies. These fields often involve complex dependencies and uncertainties, making it essential to complement theoretical knowledge with practical guidelines to ensure proper implementation.

As highlighted by studies such as G. Salvadori, De Michele, and Durante (2011) and Gräler et al. (2013), misapplications of statistical tools in practice often stem from a lack of clarity in the available literature. We see it as our responsibility to fill this gap by offering a comprehensive and well-organized framework that bridges the theoretical and practical aspects of defining multidimensional critical layers and estimating multivariate return periods. This cannot be effectively conveyed in a single paragraph without sacrificing important details that are crucial for reproducibility and understanding.

Furthermore, as this is the first part of a broader investigation, the level of detail provided here is essential to ensure that the second part (Part II) is well-supported and does not require unnecessary repetition of fundamental concepts. This structure allows for a more efficient and targeted discussion of the specific case studies and applications presented in Part II.

We hope that this explanation clarifies our perspective and illustrates the importance of the level of detail provided in this paper. We view it as a necessary step toward improving the practical application of advanced statistical methods in hydrology and addressing common sources of misinterpretation and misuse.

**Specific comments**

**Line(s) 55–56 AUTHOR(s). Vine copulas are a flexible class of dependence models consisting of bivariate building blocks. REFEREE. The Authors do not mention the problems intrinsic to modeling via Vine**

**copulas.**

Your comment raises an important point, as it allows us to delve deeper into some critical aspects of modeling with vine copulas. After reading both reviews, we identified that the conclusion of the paper could lead to misinterpretations by not explicitly addressing certain methodological limitations. Therefore, we have decided to revise the final sections of the paper. Instead of a *Discussion* and a *Conclusions* section, we have added a *Problems and Limitations* section and a *Concluding Remarks* section, where we discuss the technical challenges associated with the use of vine copulas and the strategies we have implemented to address them.

In the *Problems and Limitations* section, we will discuss the following key aspects:

1. *Computational complexity and scalability:* One of the main limitations of using vine copulas is the exponential growth in the number of parameters to be estimated as the dimensionality of the problem increases. This challenge is relevant in our context, as defining the critical layer involves handling multiple variables and conducting intensive simulations. To address this issue, we have implemented optimization techniques using Gaussian process regression (GPR), which reduces the need for exhaustive simulations. Furthermore, we plan to explore the use of parallel computing strategies and truncated vine copula models in future studies to further improve computational efficiency.

2. *Selection of the optimal structure and bivariate copula families:* The selection of the vine structure and the bivariate copula families is a critical process, as it directly impacts the accuracy of the results and the model's ability to capture extreme dependencies. We recognize that this process can be complex and computationally expensive, especially in high-dimensional applications. To address this limitation, we have implemented the Dißmann et al. (2013), which sequentially selects the optimal vine structure and copula families based on conditional dependence criteria and independence tests.

   The Dißmann algorithm optimizes the structural search by reducing the computational cost and avoiding the selection of irrelevant copulas at higher levels of the vine. This approach is particularly important in our work, where multiple hydrological variables with complex dependencies require well-defined structures. Additionally, to ensure that the selected structure properly represents the observed events, we perform a *validation of synthetic and observed events*. This process involves generating synthetic samples from the selected vine structure and comparing them with the observed data using graphical tools, such as quantile plots and joint empirical distributions. We also employ specific tests, such as the *upper tail dependence test* and *correlation test*, to verify that the model accurately captures extreme dependencies. This validation process, which is applied in the second part of our study, ensures that the selected structure is not only theoretically sound but also consistent with observed data.

   Nonetheless, we acknowledge that in more complex scenarios, future work could benefit from the development of hybrid methods that combine this algorithm with machine learning techniques to further enhance structural selection efficiency.

3. *Error propagation and simplified assumptions:* Another challenge is the potential propagation of errors in parameter estimation, especially in the presence of strong conditional dependencies. Although our approach uses crossvalidation and regularization of estimators to minimize this effect, we also recognize that, in certain applications, simplified assumptions can be problematic. Future work could explore the use of non-simplified copulas and Bayesian approaches to improve parameter estimation in highly dependent cases.

In the *Concluding Remarks* section, we will highlight that, despite these limitations, the use of vine copulas enables us to adequately capture complex dependencies among hydrological variables and accurately define the critical layer. The proposed improvements, including computational cost reduction, automated structural selection, and validation based on observed events, are important steps toward optimizing this approach and its applicability to real-world cases.

**Line(s) 58–59 AUTHOR(s). For more theoretical details, please refer to (Sklar, 1959; Nelsen, 2006). REFEREE. None of the references is pertinent to Vine copulas: more recent and relevant paper must be cited, starting from the basic one: Aas, K., Czado, C., Frigessi, A., Bakken, H. (2009). Pair-copula constructions of multiple dependence. Insurance: Mathematics and Economics, 44(2), 182–198.**

Following the reviewer's suggestion, we have incorporated more specific and updated references to further support the theoretical development of *vine copulas*. Our initial intention was for the citations provided (Sklar, 1959; Nelsen, 2006) to serve as general reference for the overall content. However, upon reviewing the text, we recognize that the phrasing of the sentence suggested that these references were directly related to *vine copulas*, which is not entirely accurate. Therefore, we will revise the sentence as follows:

*"For more theoretical details on general copulas, please refer to (Sklar, 1959; Nelsen, 2006). For theoretical details on vine copulas, please refer to (Aas et al., 2009; Czado, 2019)."*

Additionally, we will include the reference to Aas et al. (2009), as suggested by the reviewer, and add Czado (2019), one of the main sources of our research, which provides a comprehensive and practical guide to the construction and application of vine copulas. Although we had already included this reference in later sections, we recognize that it is also relevant in this context.

**Line(s) 149–151 AUTHOR(s). For instance, Kendall's is more appropriate when the joint distribution is non-Gaussian (Serinaldi, 2008). Spearman's rank correlation is based on the rankings of variable values, whereas Kendall's rank correlation assesses the concordance and discordance between pairs of observations (Czado, 2019). REFEREE. None of the references is pertinent. Much better references are the two books by Nelsen (2006)—a more theoretical one—and Salvadori et al. (2007)—a more practical one.**

As suggested, we have reviewed the relevance of the references included in this section. Serinaldi (2008) and Czado (2019) were chosen because they directly discuss the application of Kendall's and Spearman's in non-Gaussian contexts and in models involving multivariate dependence, making them relevant for our discussion. However, we recognize that the suggested references by *Nelsen (2006)* and *Salvadori et al. (2007)* provide a more comprehensive theoretical and practical perspective.

Consequently, we propose the following adjustments:

- We will retain Serinaldi (2008) and Czado (2019) due to their specific relevance to our application.

- We will add the suggested references as complementary sources to strengthen the discussion:

- Roger B. Nelsen (2006) will provide additional theoretical context on dependence measures within the framework of copulas.

- Gianfausto Salvadori et al. (2007) will enhance the practical aspect, linking the discussion to real-world hydrological applications.

The revised paragraph will read:

*Modeling the dependence between variables is fundamental for understanding and analyzing their joint behavior. To achieve this, both parametric measures, such as the Pearson correlation coefficient, and non-parametric measures, such as rank-based correlations—Kendall's  and Spearman's —are employed. Non-parametric measures are particularly favored in the estimation of dependence for compound events because the marginal distributions of these data often deviate from normality. For more theoretical background on dependence measures, see (Nelsen, 2006). For instance, Kendall's  is more appropriate when the joint distribution is non-Gaussian (Serinaldi, 2008). Spearman's rank correlation is based on the rankings of variable values, whereas Kendall's rank correlation assesses the concordance and discordance between pairs of observations (Czado, 2019; Salvadori et al., 2007).*

**Line(s) 159–ff. AUTHOR(s). Graphical tools for analyzing dependence. . . REFEREE. For intellectual honesty, the Authors should warn the reader that the interpretation of graphical results always involves a degree of subjectivity, and should always be accompanied by objective formal tests.**

This is an important observation, and we do agree that the interpretation of graphical tools for analyzing dependence, such as scatter plots, quantile-quantile plots, and dependence structure visualizations, can introduce a degree of subjectivity. To address this concern, we will include a clear recommendation in the manuscript emphasizing that graphical tools should be used as a complement to formal statistical tests.

The following revision aims to clarify the limitations of graphical tools while emphasizing their role as a complement to formal statistical analysis:

*"Graphical tools provide valuable insights into the structure of dependencies between variables, offering a visual representation that can highlight nonlinear relationships, tail dependencies, and clusters (Genest et al., 2009; Salvadori et al., 2007). However, we caution readers that the interpretation of graphical results involves a degree of subjectivity and should be complemented by formal statistical tests, such as goodness-of-fit tests, upper tail dependence tests, or correlation tests (Joe, 2015; Nelsen, 2006), to ensure robust conclusions."*

**Line(s) 175–176 AUTHOR(s). They also found that the strong bias and associated uncertainty raise doubts about the reliability of most empirical results reported in the hydrological literature. REFEREE. See also Illustration 3.18 in Salvadori et al. (2007, p. 173), where numerical experiments were carried out both on empirical and simulated data.**

As highlighted by Salvadori et al. (2007), the estimation of tail dependence coefficients $\lambda_L$ and $\lambda_U$ is particularly challenging in hydrology due to limited data

availability in extreme regions, which can lead to unstable empirical estimates. This issue is shown in Figure 3.16 of Gianfausto Salvadori et al. (2007), where the estimates become unreliable as $t \to 0^+$ or $t \to 1^-$.

To address this known limitation, we have implemented a validation procedure that combines empirical data with synthetic data generated through simulations. Following the recommendations in Salvadori et al. (2007), this approach mitigates bias and improves the robustness of our estimates by providing additional data in regions where empirical observations are scarce.

The text will be modified as follows:

*Estimating tail dependence coefficients in hydrology is prone to bias and instability due to the limited availability of extreme data (Salvadori et al., 2007). To address this issue, we validate our estimates by combining empirical data with synthetic data generated through simulations, following the recommendations in Salvadori et al. (2007), where numerical experiments showed that this combination improves the robustness of the analysis.*

**Line(s) 242 AUTHOR(s). The literature proposes various alternatives for combining multivariate analysis and non-stationarity. REFEREE. For intellectual honesty, the Authors should point out that, at present, non-stationarity is generally modeled by introducing a temporal dependence of the parameters of the marginals/copulas at play (e.g., by assuming linear and/or exponential changes of the parameters with time), but these remain mere mathematical exercises, not tested on empirical data.**

This observation brings attention to a key consideration regarding non-stationarity in the multivariate framework. While many approaches have indeed relied on introducing temporal variability in the parameters of the marginals or copulas, this is neither the only possible method nor the only one that has been empirically applied

In the univariate case, non-stationarity is often addressed by incorporating covariates into the marginal distributions rather than relying solely on explicit time-dependent parameterizations. Méndez et al. (2007) applied a non-stationary Generalized Extreme Value (GEV) model to analyze monthly extreme sea levels, explicitly considering seasonal variability and long-term trends. Later, López and Francés (2013) explored non-stationary flood frequency analysis, highlighting the influence of both climatic and anthropogenic factors on extreme event distributions. Their study emphasized the importance of integrating external covariates, such as reservoir regulation effects and climate variability indices, to improve the characterization of hydrological extremes beyond traditional stationary assumptions. More recently, Urrea Méndez and Jesus (2023) incorporated non-stationary techniques into extreme rainfall estimation, demonstrating that covariates—such as climate indices—can be used to improve probabilistic estimations without exclusively assuming parametric temporal trends.These studies show that alternative approaches exist for modeling non-stationarity beyond the simple incorporation of time-dependent parameters in traditional models.

This approach moves beyond simplistic time-based formulations, allowing for a more physically interpretable modeling of variability in extreme events multivariate context, Boumis et al. (2025) propose the use of physics-informed dynamic copulas, where dependence parameters vary not only as a function of time but also in relation to climate indices such as the Oceanic Niño Index (ONI) and the

North Atlantic Oscillation (NAO). This approach represents a more realistic alternative, as it enables the modeling of multivariate dependencies based on observable climatic factors, avoiding the abstraction of time as the sole driver of change.

To better capture these considerations, we have updated the relevant sentence as follows:

*"While much of the literature has focused on modeling non-stationarity by introducing temporal dependence in the parameters of marginal distributions or copulas (e.g., assuming linear or exponential changes over time), other approaches have also been explored. In the univariate setting, non-stationarity can be introduced through covariates in the marginal distributions (Méndez et al., 2007; López & Francés, 2013; Urrea-Méndez & del Jesus, 2023), allowing for greater flexibility in capturing long-term variability by incorporating climatic and anthropogenic influences. In the multivariate framework, alternative strategies such as physics-informed dynamic copulas have been proposed (Boumis et al., 2025), where dependence parameters evolve not only as a function of time but also in response to large-scale climate indices such as ONI or NAO, providing a physically consistent approach to modeling changing dependencies."*

**Line(s) 292–294 AUTHOR(s). This test evaluates the null hypothesis that the empirical copula comes from a specific copula;if the null hypothesis is rejected, the empirical copula does not follow the distribution of the specified copula. REFEREE. Statistically speaking, this sentence is not correct: Statistics can only offer guidance and suggestions, but never absolute truths. The words "copula does not" should be written as "copula may not".**

Your correction is well taken, and we agree that the current wording could be interpreted as overly deterministic, which is not appropriate in a statistical context. As you correctly point out, goodness-of-fit test results provide probabilistic evidence rather than absolute conclusions. Therefore, we have revised the corresponding sentence to reflect this conceptual precision.

The text has been updated to:

*"This test evaluates the null hypothesis that the empirical copula comes from a specific copula; if the null hypothesis is rejected, this suggests that the empirical copula may not follow the distribution of the specified copula."*

This revision better reflects the probabilistic nature of statistical test results and eliminates any potential misinterpretation. Additionally, this correction aligns with the general understanding that statistical tests only provide a confidence level associated with the rejection or acceptance of a hypothesis.

**Line(s) 422–ff. (3.6 Compound design events) REFEREE. The Authors are quite confused about the difference between a density and a probability distribution function: they are not the same, and they have different meanings. For instance, in the cited paper by Salvadori et al. (HESS 2011), the two different strategies proposed were based either on a probabilistic base (the Component-wise Excess one) or on a likelihood/density base (the Most Likely one). The description and the explanation given by the Authors is a superficial and confusing one, especially for practitioners. Here and elsewhere, use "most likely" instead of "most probable": in Probability Theory, a density induces a likelihood, not a probability (which, instead, is induced by a distribu-**

**tion function, i.e. the integral of a density).Line 427 refers to the Most Likely strategy outlined in the cited paper by Salvadori et al. (HESS 2011). Line 428 refers to the usage of "ensembles", as suggested in Salvadori et al. (HESS 2011).**

We are not confused, but we acknowledge that our choice of wording was influenced by our *mother tongue and the way these terms are commonly used in Spanish.* The phrase *"most probable"* was used as a *direct linguistic translation* of how we refer to density-based likelihood in Spanish. However, to ensure rigor and consistency with probability theory, we will *adopt the term "most likely"* throughout the text, aligning with standard terminology in the field.

In G. Salvadori, De Michele, and Durante (2011), two approaches are proposed for identifying representative events in a multivariate setting:

- The *Component-wise Excess Approach*, which focuses on selecting events where all variables exceed specific thresholds, prioritizing the probability of joint exceedance.

- The *Most Likely Approach*, which selects events based on the highest joint probability density, ensuring that the chosen event is the most representative in terms of likelihood.

Our methodology follows the *Most Likely Approach*, as we identify the event in the critical layer that maximizes the *joint probability density function.* This is consistent with the theoretical framework outlined in G. Salvadori, De Michele, and Durante (2011), where this method is used to determine the most typical or expected realization of an extreme event.

To avoid confusion, we will explicitly clarify that *our approach does not correspond to the Component-wise Excess strategy*, which is based on probability exceedance rather than likelihood maximization. While both strategies are valid, they serve different purposes, and our focus aligns with the density-based selection outlined in G. Salvadori, De Michele, and Durante (2011).

To improve clarity and ensure consistency with standard probability terminology, we will modify this section as follows:

*Original Title:* Most probable compound event
*Modified Title:* Most likely compound event

The revised text now states:

*"The simplest way to select the* most likely *design event among all possible options is to choose the point with the highest joint probability density. This follows the likelihood-based approach outlined by Salvadori et al. (2011)."*

These modifications aim to *eliminate any ambiguity regarding terminology*, clarify our approach, and ensure consistency with the probabilistic framework established in the literature. We appreciate the reviewer's feedback, as it has allowed us to refine this section for better clarity and methodological alignment.

**Eq. (9) is a special case of Eq. (13) in Salvadori et al. (HESS 2011), using the density f as a weight function w. Note that fXY ...W is the joint density of the distribution FXY ...W , not of the copula CXY ...W .**

We will correct and adapt the wording to properly capture the concepts. Once again, the informal way in which we speak about all these concepts has percolated

into the written paper.

**For intellectual honesty, the Authors should make it clear, and warn the reader, that there is no guarantee that the maximum found by a numerical routine will be a global maximum, rather it is very likely that it will be a local maximum, and this will be more and more likely as the dimension of the problem increases.**

A new section, *Problems and Limitations*, will be added, including a discussion on the objective behind this approach. In practice, even if the true global maximum is not found, a maximum close enough will serve, since the aim is to define an event as similar to the most likely one as possible.

We will also mention all the modern techniques used to try to improve the probability of finding the global maximum. Despite it being true that finding the global maximum in high dimensions is a tough problem, most of our machine learning and deep learning techniques depend on finding a good approximation to it, and the current development of the field indicates that the new algorithms perform well, although at a high computational cost.

**Line(s) 489–ff. (Discussion, Conclusions) REFEREE. This is not a Discussion, it is at most a replica of generic sentences already written in previous parts of the manuscript. Actually, in my opinion, in this paper there little to discuss about. The Conclusions are a collection of statements that try to justify a paper with no content.**

We do agree with the referee that the Discussion and Conclusions sections do not belong in the paper, since, without reading part II, it is difficult to provide solid conclusions to our study. We have removed both sections and instead included two new ones: *Problems and Limitations* and a *Concluding Remarks.*

The former will deal with all the topics that we have commented in this review that may render the applications of these techniques impractical or even undesirable. The latter will try to summarize the most important points to guide the reader towards the Part II of the paper.

**References**

Aas, Kjersti, Claudia Czado, Arnoldo Frigessi, and Henrik Bakken. 2009. "Pair-Copula Constructions of Multiple Dependence." *Insurance: Mathematics and Economics* 44 (2): 182–98. https://doi.org/10.1016/j.insmatheco.2007.02.001.

Boumis, Georgios, Hamed R. Moftakhari, Danhyang Lee, and Hamid Moradkhani. 2025. "In Search of Non-Stationary Dependence Between Estuarine River Discharge and Storm Surge Based on Large-Scale Climate Teleconnections." *Advances in Water Resources* 195 (January): 104858. https://doi.org/10.1016/j.advwatres.2024.104858.

Brunner, Manuela Irene. 2023. "Floods and Droughts: A Multivariate Perspective." *Hydrology and Earth System Sciences* 27 (13): 2479–97. https://doi.org/10.5194/hess-27-2479-2023.

Brunner, Manuela Irene, Jan Seibert, and Anne-Catherine Favre. 2016. "Bivariate Return Periods and Their Importance for Flood Peak and Volume Estimation." *WIREs Water* 3 (6): 819–33. https://doi.org/10.1002/wat2.1173.

Czado, Claudia. 2019. *Analyzing Dependent Data with Vine Copulas: A Practical Guide With R.* Vol. 222. Lecture Notes in Statistics. Cham: Springer International Publishing. https://doi.org/10.1007/978-3-030-13785-4.

Dißmann, J., E. C. Brechmann, C. Czado, and D. Kurowicka. 2013. "Selecting and Estimating Regular Vine Copulae and Application to Financial Returns." *Computational Statistics & Data Analysis* 59 (March): 52–69. https://doi.org/10.1016/j.csda.2012.08.010.

Gräler, B., M. J. van den Berg, S. Vandenberghe, A. Petroselli, S. Grimaldi, B. De Baets, and N. E. C. Verhoest. 2013. "Multivariate Return Periods in Hydrology: A Critical and Practical Review Focusing on Synthetic Design Hydrograph Estimation." *Hydrology and Earth System Sciences* 17 (4): 1281–96. https://doi.org/10.5194/hess-17-1281-2013.

López, J., and F. Francés. 2013. "Non-Stationary Flood Frequency Analysis in Continental Spanish Rivers, Using Climate and Reservoir Indices as External Covariates." *Hydrology and Earth System Sciences* 17 (8): 3189–3203. https://doi.org/10.5194/hess-17-3189-2013.

Méndez, Fernando J., Melisa Menéndez, Alberto Luceño, and Inigo J. Losada. 2007. "Analyzing Monthly Extreme Sea Levels with a Time-Dependent GEV Model." *Journal of Atmospheric and Oceanic Technology* 24 (5): 894–911. https://doi.org/10.1175/JTECH2009.1.

Roger B. Nelsen. 2006. *An Introduction to Copulas.* Springer Series in Statistics. New York, NY: Springer. https://doi.org/10.1007/0-387-28678-0.

Salvadori, G., C. De Michele, and F. Durante. 2011. "On the Return Period and Design in a Multivariate Framework." *Hydrology and Earth System Sciences* 15 (11): 3293–3305. https://doi.org/10.5194/hess-15-3293-2011.

Salvadori, Gianfausto, Carlo De Michele, Nathabandu T. Kottegoda, and Renzo Rosso. 2007. *Extremes in Nature: An Approach Using Copulas.* 1st ed. Water Science and Technology Library 56. Dordrecht: Kluwer academic. https://doi.org/10.1007/1-4020-4415-1.

Serinaldi, Francesco. 2008. "Analysis of Inter-Gauge Dependence by Kendall's K, Upper Tail Dependence Coefficient, and 2-Copulas with Application to Rainfall Fields." *Stochastic Environmental Research and Risk Assessment* 22 (6): 671–88. https://doi.org/10.1007/s00477-007-0176-4.

Urrea Méndez, Diego, and Manuel del Jesus. 2023. "Estimating Extreme Monthly Rainfall for Spain Using Non-Stationary Techniques." *Hydrological Sciences Journal* 68 (7): 903–19. https://doi.org/10.1080/02626667.2023.2193294.

Xu, Kui, Chenyue Wang, and Lingling Bin. 2023. "Compound Flood Models in Coastal Areas: A Review of Methods and Uncertainty Analysis." *Natural Hazards* 116 (1): 469–96. https://doi.org/10.1007/s11069-022-05683-3.

Zscheischler, Jakob, Olivia Martius, Seth Westra, Emanuele Bevacqua, Colin Raymond, Radley M. Horton, Bart van den Hurk, et al. 2020. "A Typology of Compound Weather and Climate Events." *Nature Reviews Earth & Environment* 1 (7, 7): 333–47. https://doi.org/10.1038/s43017-020-0060-z.

---

## Author Comment (AC2)

**Rebuttal of Review 2**

Dear reviewer,

We appreciate the time and effort taken to review our manuscript. Please, find below our answers to your comments.

**General Comments**

**While the manuscript primarily reviews existing concepts of return periods for compound events, it is unclear how these concepts can be generalized to all typologies of compound events. Specifically, the application of return periods to preconditioned and temporally compounding events is not well addressed. To achieve the stated goal of extending the concept to any type of compound event, the manuscript would benefit from illustrative examples that demonstrate this applicability.**

Our study follows the classification by Zscheischler et al. (2020) and emphasizes that preconditioned, multivariate, temporally compounding, and spatially compounding events can all be analyzed within a unified conceptual framework under our methodology.

While the manuscript already employs a multivariate approach that captures dependencies between different variables, we recognize the importance of explicitly clarifying how our methodology applies to each typology of compound events. To enhance clarity, we will refine the manuscript by explicitly stating that all typologies of compound events can be framed within the same conceptual approach, ensuring a unified interpretation. Additionally, we will clarify how our methodology applies to each typology described in Zscheischler et al. (2020), reinforcing that they can all be analyzed within the same methodological framework. To further support this, we will provide a detailed description of how the method is applied in each case:

1. Preconditioned events: These events are modeled by incorporating the initial system state as a variable within the dependency analysis. For example, in the case of flooding, where the occurrence of an extreme event depends on prior soil moisture conditions, this factor is explicitly included in the return period estimation through conditional copulas or probability functions. This allows for the assessment of how antecedent conditions influence the likelihood of extreme events. After the selection process, a multivariate set of events is obtained, preserving the influence of prior conditions.

2. Multivariate events: Our methodology is specifically designed to capture dependencies between multiple simultaneous extreme variables, such as the concurrent occurrence of intense precipitation and storm surges within the same system. Multivariate copulas are used to characterize these interactions and compute joint return periods, reflecting the probability of multiple

drivers occurring together. Once the selection process is completed, a multivariate set of events is obtained, maintaining the dependency structure between extreme variables.

3. Temporally compounding events: These events are handled by incorporating the relationship between extreme events occurring within a short time frame. The dependency between consecutive events is accounted for, allowing us to assess how the occurrence of one extreme event increases the probability of another occurring shortly after. This is reflected in the estimation of cumulative return periods, which consider event persistence and its impact on system recovery. At the end of the selection process, a multivariate set of events is obtained, preserving temporal dependencies and the sequential occurrence of extreme events.

4. Spatially compounding events: Our methodology enables the assessment of correlations between extreme events in different regions by incorporating spatial dependency structures. Spatial copula models are applied to estimate the joint probability of extremes occurring across multiple locations, allowing for the calculation of regional return periods that capture the simultaneous occurrence of events in different geographic areas. After the selection process, a multivariate set of events is obtained, maintaining the spatial correlation structure among the analyzed regions.

All the categories described above, end up as a collection of variables whose joint distribution needs to be characterized by means of copulas. That is, mathematically, all the typologies end up being the same problem, where copulas need to be fit and the fitted copula analyzed to derive the joint return periods. Therefore, the distinction between typologies comes from the interpretation of the results, but methodologically all the four types can be treated with the same methodology, without making any methodological different except for the event selection phase.

Additionally, we would like to highlight that our work is structured into three parts, which demonstrates the applicability of our methodological framework to different types of compound events:

- The first part, presented in this manuscript, develops the methodological framework for estimating return periods in compound events.

- The second part, addressed in a complementary study, applies this methodological framework specifically to spatially compounding events.

- The third part, although not submitted to this journal, applies the methodology to analyze multivariate events in an estuarine region, aiming to characterize flood risk considering multiple simultaneous drivers.

**Although the manuscript aims to include machine learning techniques, their description lacks detail and validation. The following areas require clarification and elaboration:**

**Page 17, Points 5C-5D: The manuscript discusses the computation of copula values in higher dimensions via integration. However, the role of Gaussian processes in this context is insufficiently explained. The algorithm is not described in detail, its numerical performance is not evaluated, and there is no validation of the method.**

Gaussian Processes are used as a universal approximator or emulator of the function that we want to evaluate. Since computing values of a multivariate cumulative distribution function (CDF) is a costly endeavor, the Gaussian Processes serve to approximate such a non-linear function with less computational cost.

To improve the clarity of the manuscript, we will include a more detailed methodological description of Gaussian Process Regression (GPR) in the supplementary information. This will provide a better understanding of the role of GPR in our methodology and its applicability in estimating computationally expensive functions.

The use of GPR has been extensively studied and applied in the approximation of high-cost computational functions in various contexts. For example, Ba and Joseph (2012) developed a composite Gaussian process model to emulate computationally expensive functions, capturing both global trends and local details.

Additionally, Zhuang et al. (2025) applied GPR in the context of high-dimensional American option pricing, highlighting its potential to mitigate computational challenges in high-dimensional settings. Their study demonstrates that GPR-based approaches can be adapted to efficiently model complex systems without a significant increase in computational cost, which aligns with our use of GPR to approximate the Vine copula CDF while maintaining computational efficiency.

These studies are examples of the effectiveness of Gaussian processes in approximating computationally expensive functions, supporting their application in our study. We believe this addition will clarify the role of GPR within our methodological framework and provide a stronger justification for its use.

The validation and calibration performance is carried out in Part II of the paper, since this first part is a detailed description of the methodology, presenting potential variations and citing all the relevant sources that we know about.

**Page 18: The determination of the highest probability density point in the critical layer is attributed to "computational optimization techniques," but these techniques are neither named nor described.**

Once the critical layer has been defined, there are two potential ways forward to define events of interest. The first one is to determine the most likely event from the critical layer. To determine this event, we have used the MLE (Maximum Likelihood Estimation) algorithm of the Spotpy Python library, which -in spite of its name- does not maximize the statistical likelihood, but it does a classical gradient descent optimization -considering the derivatives of the function-. This procedure works well for up to three dimensions, because above that number there is a high probability of it getting stuck in a local maximum.

The alternative strategy is to not look for the most likely event, but simply to sample events from the critical layer proportionally to their likelihood. However, this procedure faces two complications. First, there is a need to compute the value of the CDF at each point, which implies a multidimensional numerical integration, which is highly time consuming. Second, to sample from the critical layer we need the CDF for all the events at the layer, which is difficult to calculate accurately.

To remediate the first problem, we use Gaussian Process Regression (GPR) as a non-linear approximator to the real value of the multivariate CDF. We compute the value of the CDF using Monte Carlo integration methods for a number of points that serve as interpolation basis for GPR. Then, this methods provides approximated values of the CDF, speeding up the process. The number of points

of the interpolation base serve to control the approximation error to any desired level.

The second problem is dealt with using the Metropolis-Hastings algorithm, which allows us to sample from complicated probability distributions where the CDF cannot be computed. Using an auxiliary probability distribution, this method allows us to generate a sample that grows iteratively, which, in the end, converges to the desired unknown distribution.

Sampling for the critical layer in this way, allows us to generate a collection of events from which the most likely one -or an event almost indistinguishable from the most likely one- can be obtained. Even more, since a complete collection of events is obtained, multiple events related to the same return period can be analyzed and see how the effect of such events may differ in the impact variable.

To improve the clarity of the methodology presented in the manuscript, we will incorporate the following description in the methodological section:

*The identification of the most probable design event within the critical layer was performed following two complementary approaches, depending on the dimensionality of the problem.*

*First, for cases involving up to three dimensions, we employed Maximum Likelihood Estimation (MLE) as implemented in Spotpy (Spotpy, 2024). Contrary to its name, this algorithm does not maximize the statistical likelihood but rather applies a classical gradient-descent optimization, leveraging function derivatives to iteratively refine the estimate of the most probable event. This method is computationally efficient; however, for problems beyond three dimensions, it presents a high risk of converging to local maxima, reducing its effectiveness.*

*As an alternative, when dealing with higher-dimensional spaces, we employed a sampling-based strategy that does not rely on direct maximization but instead proportionally samples events from the critical layer based on their likelihood. This approach, however, faces two main computational challenges:*

1. *The need to evaluate the cumulative distribution function (CDF) at multiple points, requiring multidimensional numerical integration, which is computationally expensive.*

2. *The necessity to estimate the CDF for all events within the critical layer, which is challenging to compute accurately.*

*To address the first issue, we leveraged Gaussian Process Regression (GPR) as a surrogate model to approximate the true multivariate CDF. We computed reference CDF values using Monte Carlo integration over a selected set of interpolation points, which were then used to train the GPR model. This approach significantly reduces computational costs while maintaining control over approximation errors.*

*For the second issue, we utilized the Metropolis-Hastings algorithm, which allows us to efficiently sample from complex probability distributions where direct CDF computations are infeasible. By iteratively refining an auxiliary probability distribution, this method generates a representative sample of events that converge to the true distribution of extreme events in the critical region.*

*This refined sampling process allows us to obtain a collection of extreme events, from which we can extract the most probable one—or an event nearly indistinguishable from it. Furthermore, this method enables the analysis of multiple events with*

*the same return period, providing deeper insight into how different realizations of extreme events may impact the target variable.*

*The complete implementation of this methodology, including its application in high-dimensional contexts, is detailed in Part II of this study, where we validate its effectiveness and illustrate its practical relevance in extreme event characterization.*

We believe that these modifications will contribute to a better integration between Part I (methodological framework) and Part II (case study) while enhancing clarity in the presentation of the optimization process. We sincerely appreciate the reviewer's observation, which has allowed us to strengthen the exposition of our methodology.

**Section 3.6.2: The manuscript references the use of the Metropolis-Hastings algorithm. However, no algorithm details are provided, and no numerical validation or examples are included.**

We propose modifying Section 3.6.2 of the manuscript to include a more detailed description of the Metropolis-Hastings algorithm, which was previously mentioned in line 475 of Part I of the article. However, we acknowledge that the original manuscript did not include the proper reference to the foundational work on the method. This will be corrected by citing Hastings (1970), who formalized the algorithm, along with the original formulation by Metropolis et al. (1953). In addition to this correction, we will clarify its implementation and provide numerical evidence to support its functionality. The suggested wording for the revised manuscript is as follows:

*"To obtain a representative sample of the probability distribution within the critical layer, the Metropolis-Hastings algorithm is implemented (Metropolis et al., 1953; Hastings, 1970). The process begins with the selection of an initial event within the critical layer, from which a sequence of events is generated using an auxiliary proposal distribution. In each iteration, a new candidate event is proposed, and its probability density is compared to that of the current event. If the candidate has a higher density, it is automatically accepted; otherwise, it is accepted with a probability proportional to the ratio of both densities. This iterative procedure allows the construction of a sample that, after a sufficient number of iterations, converges to the target distribution, ensuring an adequate representation of the variability in extreme events.*

*In Part II of this study, numerical validation is presented to demonstrate the effectiveness of the method in generating representative samples of the probability distribution. This enables both the identification of the event with the highest density and the analysis of multiple events associated with the same return period."*

With this modification, we aim to improve the methodological clarity of the process, particularly by emphasizing the selection of the proposal distribution, as it plays a key role in the efficiency of the sampling procedure and the convergence to the target distribution. This revision provides a clearer understanding of the role of Metropolis-Hastings in our methodology.

**In summary, while the manuscript provides a comprehensive review of return period concepts, further clarification and illustrative examples are needed to demonstrate the applicability of these concepts to diverse types of compound events. Additionally, the integration of machine learning techniques should be substantiated with detailed methodologies and numerical validations.**

We appreciate the reviewer's comment and agree that including additional illustrative examples can improve the understanding of the applicability of the concepts presented in the manuscript. We will revise the text to incorporate more examples where possible and relevant.

However, we would like to emphasize that this manuscript is the first part of a three-part study. In particular, the second part of this study, which is currently under review, focuses on applying the methodology to spatially and temporally compounding events, providing detailed numerical validations and case studies. Additionally, there is a third part of the study, which, although not submitted to this journal, applies the same methodological framework to multivariate compound events in estuarine environments, considering river discharge, wind, and marine/coastal variables to analyze interactions between different extreme phenomena affecting estuarine dynamics.

Since this manuscript primarily focuses on methodological formulation, most of the validation and practical applications are presented in the second part of the study.

**Specific comments**

**Section 3.1.2: It is unclear to me the difference between parametric and non-parametric measures of dependence. I guess that the distinction is between rank-invariant measures and other measures like Pearson's correlation. Also the intuitive distinction between Spearman's rho and Kendall's tau at line 150 should be better explained.**

We appreciate the reviewer's insightful comment and agree that a more detailed explanation of these concepts will enhance the manuscript's clarity. Differences between Parametric and Non-Parametric Dependence Measures:

- Parametric Measures: These measures, such as Pearson's correlation coefficient, assume a linear relationship between variables and require that the data follow a normal distribution. They are sensitive to outliers and may not effectively capture non-linear relationships.

  Non-Parametric Measures: These include coefficients like Spearman's rho and Kendall's tau, which do not assume a specific distribution and are based on the ranks of the data rather than their actual values. This makes them more robust to non-normal distributions and outliers, allowing them to capture monotonic relationships that are not necessarily linear.

Differences between Spearman's Rho and Kendall's Tau:

- Spearman's Rho: Calculated as the Pearson correlation coefficient between the ranked variables, Spearman's rho assesses how well the relationship between two variables can be described using a monotonic function. It is sensitive to differences in ranks and can be influenced by the presence of tied ranks.

- Kendall's Tau: This coefficient measures the difference between the probability of observing concordant versus discordant pairs in the data (Okoye and Hosseini 2024). It is considered more robust in the presence of ties and provides a more direct probabilistic interpretation of the strength of association between two variables.

To address this observation in the manuscript, we will implement the following revisions:

1. Enhance the explanation in Section 3.1.2 by explicitly differentiating between parametric and non-parametric dependence measures, supported by the relevant references.

2. Provide a clearer intuitive explanation around line 150 regarding the conceptual differences between Spearman's rho and Kendall's tau, highlighting their respective applications and limitations, and substantiating with appropriate citations.

**Section 3.2: please, notice that AIC is not a test, but a selection criterion.**

We have corrected the wording to make this point more clear.

**Eq. (1): It is unclear what is x and what is R_i(x).**

We appreciate the reviewer's observation and acknowledge that an adjustment in the mathematical notation is necessary to enhance clarity and ensure a precise interpretation of the expressions used.

In this context, $R(x)$ represents the rank of the observation $x$ within the dataset. As described in Section 3B of the manuscript, to compute the empirical copula, it is common practice to transform the original variables into the standard uniform space ($[0, 1]$). This transformation is performed using the expression:

$$\hat{F}(x) := \frac{R(x)}{n + 1}$$

where $R(x)$ is the rank of $x$ among all observations for a given variable, and $n$ is the total number of observations. By applying this transformation, the original variables $(X, Y, ..., W)$ are converted into pseudo-observations $(U_X, U_Y, ..., U_W)$, which follow a uniform distribution in ($[0, 1]$).

To address this observation in the manuscript, we will implement the following revisions:

1. Clarify the definition of $R(x)$ in the text, explicitly explaining its role in the variable transformation for empirical copula construction.

2. Include a brief note in the corresponding equation, ensuring that readers understand that $R(x)$ represents the rank of the observation, rather than an additional function.

   These modifications will improve the clarity of the manuscript and facilitate the interpretation of the notation used.

**Eq. (3) and (4): It is difficult to understand in which sense X and Y are conditional events rather than random variables. Analogously, F_X(X) should be a random variable and not a conditional event.**

We acknowledge the reviewer's comment and recognize that the notation in Equations (3) and (4) was not presented clearly. Specifically, the way $X$ and $Y$ were defined may have led to the incorrect interpretation that they represent conditional events rather than random variables.

In our approach, $X$ and $Y$ are continuous random variables, while $U = F_X(X)$ and $V = F_Y(Y)$ correspond to their transformation via the probability integral transform. This process allows any continuous random variable to be mapped into the uniform space $[0, 1]$ and is widely used in copula modeling to represent dependence structures without imposing assumptions on the marginal distributions.

To improve clarity in the manuscript and avoid any possible ambiguities, we will implement the following modifications:

1. Clarify in the text that $X$ and $Y$ are random variables, while $U$ and $V$ result from applying the probability integral transform, ensuring they follow a uniform distribution in $[0, 1]$.
2. Reword the explanation of Equations (3) and (4) to prevent any misinterpretation regarding the term "conditional events."

We believe these adjustments will enhance the clarity of the manuscript and ensure that the interpretation of Equations (3) and (4) is well understood. We sincerely appreciate the reviewer's insightful suggestion, which will undoubtedly contribute to improving the presentation of our work.

**Eq. (5): W is not the copula function but the random variable defined by C(U,V).**

In this context, $W = C(U, V)$ is a univariate random variable and not the copula function itself. Equation (5) defines the Kendall distribution function, which is expressed as:

$$K_c(t) = \Pr[W \leq t] = \Pr[C(U, V) \leq t]$$

where $W$ represents a scalar value derived from the copula function, and $t$ is a probability threshold that separates the supercritical and non-critical regions.

To improve clarity in the manuscript and avoid any misinterpretation, we will implement the following adjustment:

1. Explicitly state in the text that $W$ is a univariate random variable derived from the copula and not the copula function itself.

We believe this modification will enhance the understanding of the role of $W$ in Equation (5).

**Sections 4 and 5 are quite similar. They should be merged and their scope should be better defined.**

We do agree with the referee that the Discussion and Conclusions sections do not belong in the paper, since, without reading Part II, it is difficult to provide solid conclusions to our study. We have removed both sections and instead included two new ones: *Problems and Limitations* and *Concluding Remarks.*

The former will address all the aspects discussed throughout this review that may challenge the practical implementation of these techniques or even render them unsuitable in certain scenarios. The latter will summarize the key points and guide the reader towards Part II of the paper, where we will present a practical application of the methodology introduced in this study. This second part will demonstrate the applicability of our approach using real-world data and case studies, further supporting its robustness and relevance.

**References**

Ba, Shan, and V. Roshan Joseph. 2012. "Composite Gaussian Process Models for Emulating Expensive Functions." *The Annals of Applied Statistics* 6 (4): 1838–60. https://doi.org/10.1214/12-AOAS570.

Okoye, Kingsley, and Samira Hosseini. 2024. "Correlation Tests in R: Pearson Cor, Kendall's Tau, and Spearman's Rho." In *R Programming: Statistical Data Analysis in Research*, edited by Kingsley Okoye and Samira Hosseini, 247–77. Singapore: Springer Nature. https://doi.org/10.1007/978-981-97-3385-9_12.

Zhuang, Jirong, Deng Ding, Weiguo Lu, Xuan Wu, and Gangnan Yuan. 2025. "A Gaussian Process Based Method with Deep Kernel Learning for Pricing High-Dimensional American Options." *Computational Economics*, January. https://doi.org/10.1007/s10614-024-10833-9.